# MindCustomer: Multi-Context Image Generation Blended with Brain Signal

**Muzhou Yu** [* † 1] **Shuyun Lin** [* 2] **Lei Ma** [3 4] **Bo Lei** [3] **Kaisheng Ma** [2]

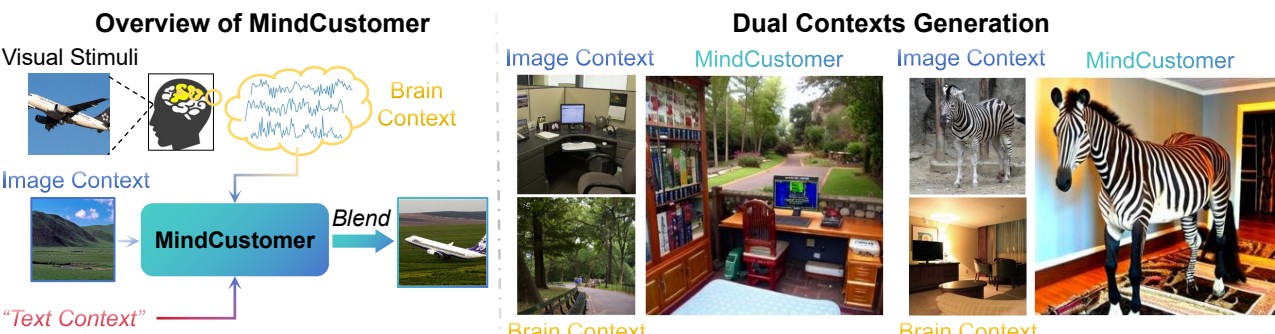

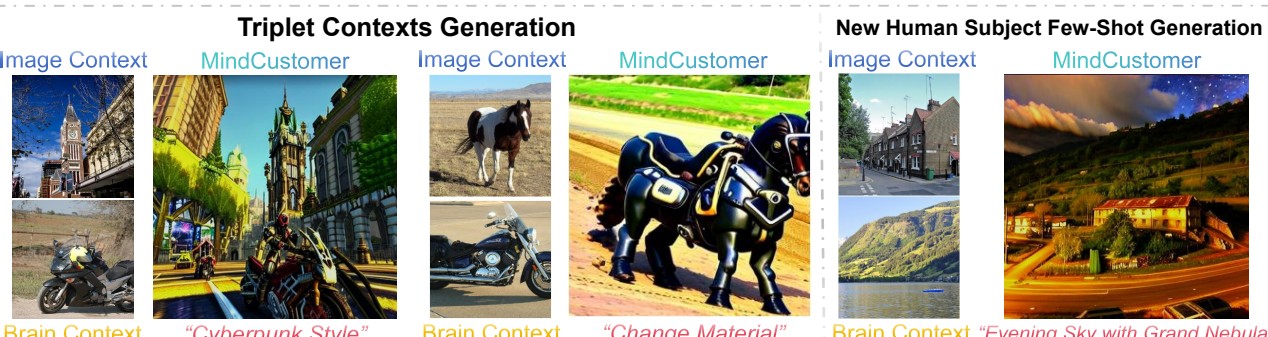

*Figure 1.* MindCustomer seamlessly integrates visual brain signals into multi-context image generation. Given the scarcity of real-world brain data, it efficiently performs real-time single-image generation without additional inputs (e.g. masks or muti-views of the same image context). Text context is optional, which provides more semantics or control over the image and brain contexts.

## Abstract

Advancements in generative models have promoted text- and image-based multi-context image generation. Brain signals, offering a direct representation of user intent, present new opportunities for image customization. However, it faces challenges in brain interpretation, cross-modal context fusion and retention. In this paper, we present MindCustomer to explore the blending of visual brain signals in multi-context image generation. We first design shared neural data augmentation for stable cross-subject brain embedding by introducing the Image-Brain Translator (IBT) to generate brain responses from visual images. Then, we propose an effective cross-modal information fusion pipeline that mask-freely adapts distinct semantics from image and brain contexts within a diffusion model. It resolves semantic conflicts for context preservation and enables harmonious context integration. During the fusion pipeline, we further utilize the IBT to transfer image context to the brain representation to mitigate the cross-modal disparity. MindCustomer enables cross-subject generation, delivering unified, high-quality, and natural image outputs. More-

*Equal contribution †Work done during internship at Beijing Academy of Artificial Intelligence. [1]Xi'an Jiaotong University [2]Tsinghua University [3]Beijing Academy of Artificial Intelligence [4]Peking University. Correspondence to: Lei Ma <lei.ma@pku.edu.cn>, Bo Lei <b.lei.2022@hotmail.com>, Kaisheng Ma <kaisheng@mail.tsinghua.edu.cn>.

*Proceedings of the $42^{nd}$ International Conference on Machine Learning*, Vancouver, Canada. PMLR 267, 2025. Copyright 2025 by the author(s).

over, it exhibits strong generalization for new subjects via few-shot learning, indicating the potential for practical application. As the first work for multi-context blending with brain signal, MindCustomer lays a foundational exploration and inspiration for future brain-controlled generative technologies.

## 1. Introduction

With the advance of generative models (Chang et al., 2023; Ramesh et al., 2022a), text-to-image generation has garnered increasing attention. Users can achieve a desired visual effect or convey specific information by providing diverse text prompts. More recently, advancements in large-scale models, such as diffusion (Rombach et al., 2022; Xu et al., 2023), CLIP (Radford et al., 2021b), and large language models (Naveed et al., 2023; OpenAI, 2023) have further enhanced image generation(Kawar et al., 2023; Nichol et al., 2022). Based on this, numerous studies (Ding et al., 2024; Kumari et al., 2023; Ruiz et al., 2023; Gal et al., 2023c) have made significant strides in multi-context blended image generation from image and text prompts. It enables high-quality and customized creations with greater control, which has broad applications, including art creation, user-specific design, and virtual interactions.

Simultaneously, we notice that research on brain signal interpretation leveraging decoding technology enhances the understanding of biological information (Gong et al., 2024; Gao et al., 2024). However, current works are mainly limited to the brain signal reconstruction (Scotti et al., 2024; Wang et al., 2024; Xia et al., 2024). As a significant semantic modality, brain signals straightforwardly reflect users' personalized thoughts and preferences. We believe that it has a broader extension to customized image creation. Compared to traditional prompts from text and image modalities, brain signals offer a direct representation of user intent, paving the way to produce more efficient and personalized generation. Based on the above observation, this paper pioneers the exploration of multi-context image generation blended with brain signals. It integrates cross-modal contexts from the image, text, and brain signal into a unified generative model to facilitate image customization.

However, it lays several key challenges. (1) Cross-subject brain encoding: to achieve unified brain context extraction across different subjects, it is crucial to precisely capture semantic brain embeddings under the shared data scarcity. (2) Multi-modal integration: the generative model should integrate implicit representations from multiple modalities in a cohesive manner; otherwise, the embeddings from different modalities may confuse the model's latent space. (3) Multi-context preservation: when multiple contexts are fused in one generative model, it is important to ensure that the generated images retain the semantics originating from each context, without being overshadowed or altered.

In this paper, we present MindCustomer to address this challenging task. We incorporate the visually evoked brain signals across different human subjects, rich in diverse semantic information from viewing natural images, as the brain contexts. MindCustomer leverages the diffusion model (Xu et al., 2023) as the base structure for prompt-guided image generation to synthesize high-quality images. More importantly, we identify three critical points: (1) We introduce the Image-Brain Translator (IBT) for cross-subject brain data augmentation, ensuring subsequent stable embeddings. Trained to generate pseudo-brain data from image inputs, IBT works alongside a semantic mapper to precisely capture semantic contexts in brain embeddings. (2) We propose an effective cross-modal information fusion pipeline for multi-context preservation during diffusion generation. In order to blend different contexts, we design diffusion fine-tuning based on image context to learn the image semantics, and then we freeze the fine-tuned diffusion model and lightly optimize the brain context with the target of the image context. This helps mitigate semantic conflicts between different contexts. (3) Besides, during the above process, we further utilize the IBT and brain embedding model to transfer image context to brain-embedding space to mitigate the representation disparity across modalities. Thus, the above-designed brain-blended cross-modal fusion enables final natural integration and semantic preservation across modalities, producing natural and high-quality blending results. MindCustomer performs real-time generation on a single image in light of the scarcity of neural data. Meanwhile, it efficiently achieves multi-context blending without additional masks or samples.

As the first work, MindCustomer successfully implements multi-context blended with brain signal in image generation. It achieves high-quality image customization guided by cross-subject brain contexts. Furthermore, in real-world scenarios where data from new subjects is often limited, we apply MindCustomer with few-shot learning to showcase its generalization capability. Figure 1 illustrates the overview and performance of MindCustomer.

In summary, our contributions are:

- We explore the task of blending brain semantics in multi-context image generation for the first time, and present the effective MindCustomer to achieve it.

- We design the shared neural data augmentation via IBT to ensure stable cross-subject brain encoding. Furthermore, we present an effective pipeline to preserve and fuse contexts naturally during generation, alongside IBT to mitigate the cross-modal disparity.

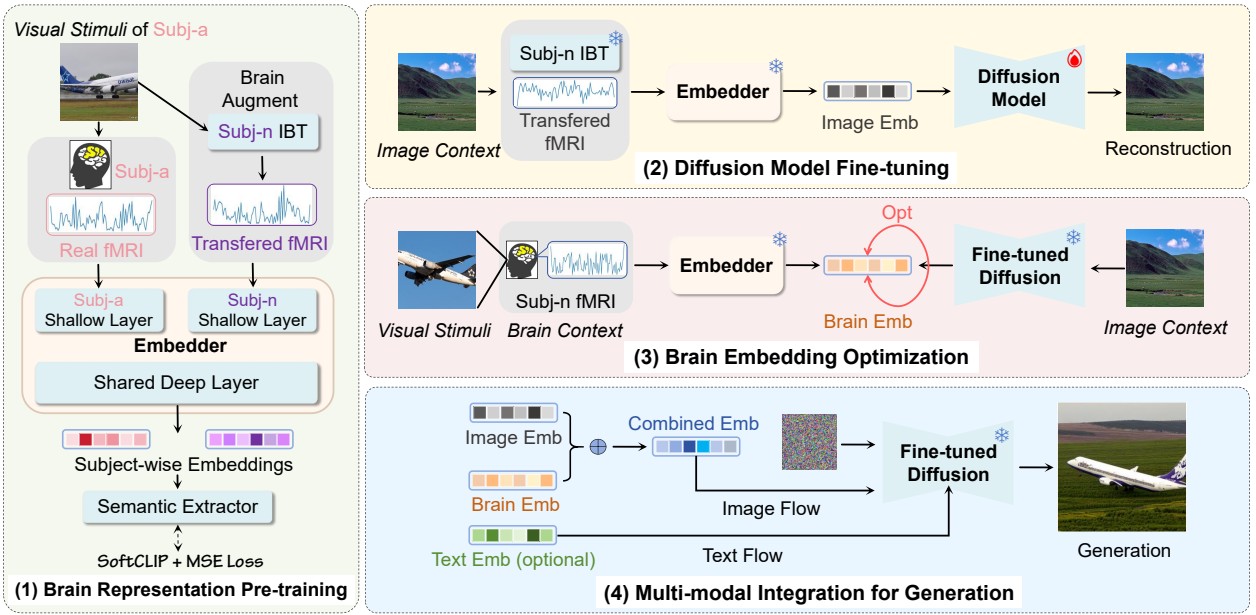

*Figure 2.* Pipeline of MindCustomer. (1) IBT simulates shared brain data across human subjects to augment brain embedding and the semantic extractor is designed to refine the representation. (2) We transfer the image context to the brain embedding space and utilize it for the diffusion fine-tuning. (3) The brain context is optimized by using the fine-tuned diffusion, further ensuring cross-context adaptation. (4) We combine these multi-modal embeddings to a new one, and finally generate the coherent blending results.

- MindCustomer achieves high-quality and seamless image generation guided by brain semantics. It can further generalize to new subjects, performing strong real-world application capability.

## 2. Related Work

Advancements in diffusion models (Ho et al., 2020; Dhariwal & Nichol, 2021; Song et al., 2021) have sparked interest in personalized image creation, allowing individuals to generate images based on their creative intent. Many works adopt fine-tuning diffusion as a core principle (Ruiz et al., 2023; Gal et al., 2023a). DreamBooth (Ruiz et al., 2023) fine-tunes the diffusion model to associate a unique identifier with a target subject, utilizing a prior preservation loss to maintain model generalization in few-shot tuning. Textual Inversion (Gal et al., 2023a) leverages a text embedding that is learnable for encoding a subject concept. Blip-Diffusion (Li et al., 2023a), building on BLIP-2's approach (Li et al., 2023b), pre-trains a multimodal encoder to generate visual representations aligned with text prompts, enabling a diffusion model to leverage these representations for learning subject-specific features and creating novel renditions. Recent research also explores merging multiple concepts for generation (Avrahami et al., 2023; Tewel et al., 2023a; Gu et al., 2023a; Liu et al., 2023b;c). Custom Diffusion (Kumari et al., 2023) implements this by employing a constrained optimization method with a closure solution.

FreeCustom (Ding et al., 2024) designs a weighted mask strategy to fuse input concepts with more focus. Overall, these advancements in image creation facilitate user-driven image generation guided by text and image prompts.

The latest developments in hardware have allowed for the recording and analysis of various brain signals in neuroscience labs (Sun et al., 2024; Taheri et al., 1994; Hari & Lounasmaa, 1989). Brain signals, rich in semantic information, are crucial for studying neural mechanisms (Daly & Wolpaw, 2008; Chaudhary et al., 2016; Weiskopf, 2012). Decoding visual brain signals has become a key research area, helping researchers understand human visual perception (Cox & Savoy, 2003; Horikawa & Kamitani, 2015; Schoenmakers et al., 2013; VanRullen & Reddy, 2018; Gu et al., 2023b; Quan et al., 2024). The development of generative models like diffusion has significantly advanced high-quality brain signal decoding (Wang et al., 2024; Xia et al., 2024; Scotti et al., 2023). A predominant approach (Scotti et al., 2024; Shen et al., 2024) involves aligning brain signal encodings with the CLIP space (Radford et al., 2021b), which serves as a condition input for generative models to perform the reconstruction. Additionally, Takagi et al. (Takagi & Nishimoto, 2023) reconstruct high-resolution images from brain activity while preserving rich semantic details, without the need for training or fine-tuning complex deep generative models. Mind-Vis (Chen et al., 2023) utilizes Masked Autoencoders (MAE) to enhance brain signal encoding, enabling more accurate reconstructions. Given

the variability of brain signals across individuals, cross-subject decoding methods have been proposed (Wang et al., 2024; Xia et al., 2024) to integrate the encodings and reconstructions of multiple subjects within a unified model. Moreover, some works extend brain signals to more tasks (Xia et al., 2024; Shen et al., 2024; Chen et al., 2024a), such as visual grounding, visual-question-answer, and stylized reconstruction. As a concurrent work, MindPainter (Yu et al., 2025) uses the brain signal as a condition for masked image painting via self-supervised learning. Different from these works, this paper presents a mask-free, cross-subject, and multi-context (modalities in image, text, and brain) blending task for image generation. We design the shared neural data augmentation and cross-modal information fusing pipeline to effectively achieve high-quality image blending results. MindCustomer further broadens the way for brain-controlled image creation.

## 3. Method

### 3.1. Preliminaries

**NSD Data** To better understand the brain-driven context, we first introduce the data related to visual brain signals. In this paper, we utilize the widely-used data, Natural Scenes Dataset (NSD) (Allen, 2022), as our brain contexts. This dataset includes high-resolution 7-Tesla fMRI scans from eight healthy adult participants, who were tasked with viewing various natural images sourced from the MS-COCO dataset (Lin et al., 2014). In accordance with standard practices (Scotti et al., 2024; Wang et al., 2024; Takagi & Nishimoto, 2023; Shen et al., 2024; Chen et al., 2024a), we utilize all subject-wise data from four subjects (Subj1, 2, 5, 7) as the training data. Each subject viewed 8859 individual images. And the remaining data of 982 images were viewed by all four subjects as the test data.

**Diffusion Model** Diffusion models (Rombach et al., 2022; Xu et al., 2023) have shown remarkable efficacy in synthesizing high-quality images. Leveraging extensive text-image pair datasets during training, they are capable of generating highly accurate visual representations from textual input. In this study, we leverage versatile diffusion (VD) (Xu et al., 2023), which is a multi-modal diffusion with independent flows for image and text prompts. This inherently enables us to condition the model with both image and text modalities, and we mainly focus on integrating visual brain signals with image modality in VD to generate high-quality images. We initialize our model with the publicly pre-trained VD, and the subsequent fine-tuning process is built upon this prior.

### 3.2. MindCustomer

Given image contexts, brain signals from different human subjects and text prompts, our target is to perform mask-free blending to generate images. The generation results should

naturally blend with the provided contexts, meanwhile preserving the relevant semantic characteristics. We propose MindCustomer to achieve this task. Considering the limited availability of brain data from different subjects, it is difficult to construct large, labeled datasets for training generative models. In this work, we perform efficient real-time generation on a single image, which holds greater practical significance for real-world applications. MindCustomer consists of four parts, which is illustrated in Figure 2.

**Brain Representation Pre-training** To tackle the inconsistency for cross-subject brain embedding, inspired from (Scotti et al., 2024; Wang et al., 2024), we apply individual-to-shared encoding to encode fMRI voxels into the universal latent space of the pre-trained CLIP model. However, previous methods only allow the subject-wise data for training, where the limited data affects the encoding capability. We introduce Image-Brain Translator $\mathcal{G}_\eta$ (IBT) to generate pseudo voxels to augment the training data without manual collecting. $\eta$ refers to the parameters of the model $\mathcal{G}$. IBT is trained to simulate subject-specific brain signals from a given input image. During the brain representation pre-training stage, we concurrently feed fMRI data from multiple subjects (including true and generated fMRI) in response to the same stimulus into a shared encoding model. This approach effectively augments the fMRI training dataset, as the stimulus presented to each subject differs from those in the original dataset. As a result, we can generate shared brain representations across subjects, rather than treating each subject's data as isolated samples, thereby enhancing the stability of the embedding training data. Specifically, we utilize the pairs of fMRI voxels $B$ and corresponding visual stimuli $I \in \mathbb{R}^{N \times 256 \times 256}$ in a batch size of $N$ to train the subject-wise IBT. We adopt the adaptive max pooling (Wang et al., 2024) to achieve the subject-invariant voxel sizes. Thus, we construct the fixed size of ground truth voxel $B'$ from (13000~16000) to 8192. The training function of IBT $\mathcal{G}_\eta$ can be formulated as:

$$
\begin{aligned}
E_I &= \text{CLIP}_{\text{Image}}(I), \\
\mathcal{L}_{IBT} &= L_{\text{MSE}}(\mathcal{G}(E_I, \eta), B'),
\end{aligned} \tag{1}
$$

We feed the $I$ into the pre-trained CLIP ViT/L-14 (Radford et al., 2021b) to obtain the latent feature $E_I$ of size $N \times 1024$. $L_{MSE}$ is the MSE loss function. Note that the CLIP model remains frozen.

Next, we elaborate on the pre-training of cross-subject brain embedder. We denote different subjects as $a$, $b$, $c$, and $d$. For a pair of $(I_a, B_a)$ from subject $a$, we first obtain the text caption $T_a$ from image $I_a$, and feed both $I_a$ and $T_a$ into CLIP model to get the corresponding image and text CLIP embedding $E_{Ia}$ and $E_{Ta}$. Then we fix the size of $B_a$ to $B'_a \in \mathbb{R}^{8192}$, and apply the IBT to generate $B'_b \in \mathbb{R}^{8192}$, $B'_c \in \mathbb{R}^{8192}$ and $B'_d \in \mathbb{R}^{8192}$ from input $I_a$. We employ the shallow subject-wise em-

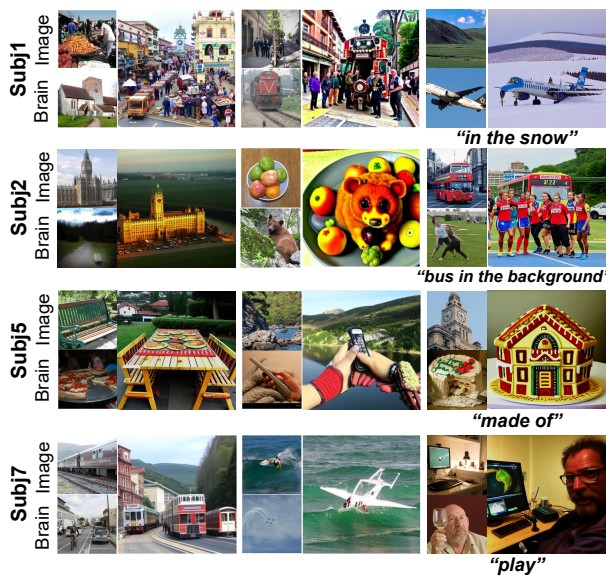

*Figure 3.* Multi-context generation results across different subjects.

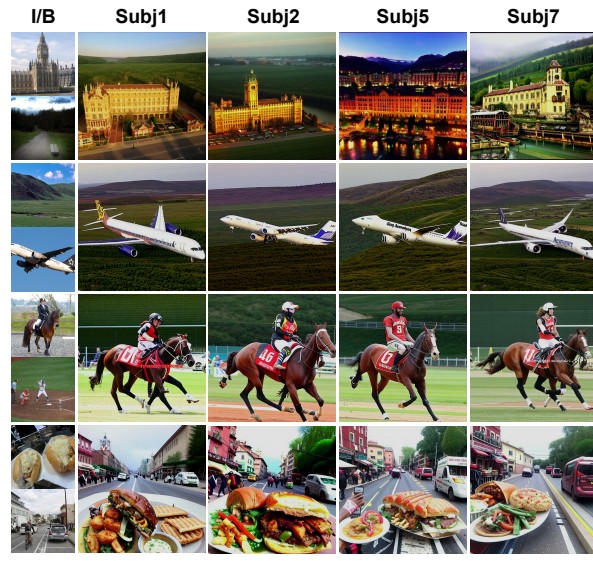

*Figure 4.* We utilize the same image context (I) and visual stimuli (B) for different subjects to blend. It demonstrates the robustness of MindCustomer and individual variability in generation.

bedders $\mathcal{E}_f$ and deep shared embedders $\mathcal{E}_\Gamma$ to output embeddings of voxels, $(e_{Ia} \in \mathbb{R}^{8192}, e_{Ta} \in \mathbb{R}^{8192})$, $(e_{Ib}, e_{Tb})$, $(e_{Ic}, e_{Tc})$, and $(e_{Id}, e_{Td})$, which will be supervised by CLIP-encoded text and image. Further, we utilize the SoftCLIP loss and MSE loss to optimize the error between the predicted embeddings and corresponding CLIP embeddings $\mathcal{L}_{\text{emb}} = \mathcal{L}_{\text{SoftCLIP}} + \mathcal{L}_{\text{MSE}}$, where

$$\mathcal{L}_{\text{SoftCLIP}} = -\sum_{i=1}^{N}\sum_{j=1}^{N}\left[\log\frac{\exp(E_i \cdot E_j/\tau)}{\sum_{n=1}^{N}\exp(E_i \cdot E_n/\tau)} + \log\frac{\exp(e_i \cdot E_j/\tau)}{\sum_{n=1}^{N}\exp(e_i \cdot E_n/\tau)}\right] \tag{2}$$

$$\mathcal{L}_{\text{MSE}} = \frac{1}{N}\sum_{i=1}^{N}(e_i - E_i)^2, \tag{3}$$

With the pseudo voxel augmentation and pre-training process, we can embed the cross-subject voxels into latent CLIP embedding, which benefits accurate semantic extraction for diffusion-based generation. Furthermore, to extract more semantically enhanced features, we introduce the mapper from ClipCap (Mokady et al., 2021) to guide our embedding model to explore the text semantics from brain signals. Since the mapper is formerly trained for image caption, it effectively transfers the vision features to language features, we utilize it as the supervisor for enhanced brain embeddings. We design the training of semantic extractor $\mathcal{M}_\sigma$ for the embeddings of voxels. The output features of the extractor $\mathcal{M}_\sigma$ are also aligned with the loss of SoftCLIP and MSE.

$$\mathcal{L}_{\text{SoftCLIP}} = -\sum_{i=1}^{N}\sum_{j=1}^{N}\left[\log\frac{\exp(S_i \cdot S_j/\tau)}{\sum_{n=1}^{N}\exp(S_i \cdot S_n/\tau)} + \log\frac{\exp(s_i \cdot S_j/\tau)}{\sum_{n=1}^{N}\exp(s_i \cdot S_n/\tau)}\right] \tag{4}$$

$$\mathcal{L}_{\text{MSE}} = \frac{1}{N}\sum_{i=1}^{N}(s_i - S_i)^2, \tag{5}$$

$S$ is the feature of the ground truth mapper, and $s$ is the predicted feature of the extractor $\mathcal{M}_\sigma$. Note that the IBTs, shallow embedders, deep embedders, and semantic extractor are all MLP-based models. Especially, the IBT has only 3 layers of MLPs, which are light for efficient training.

**Diffusion Fine-tuning with Embedding Optimization** We observed that when directly fusing brain modality with image modality, the semantic gap between the modalities often results in inferior generation outcomes. Although alignment within the CLIP latent space has been attempted, the implicit and complex representations of brain signals still exhibit semantic discrepancies with the image space. Therefore, we propose to apply the IBT $\mathcal{G}_\eta$ to transform the image context $I$ into brain voxel, aiming to eliminate inter-modal inconsistencies and reduce the semantic bias. We feed the transferred image context $B_p$ into brain embedder $\mathcal{E}$ to obtain the embeddings $e_p$ and fine-tune the versatile diffusion to reconstruct the image. It not only allows the diffusion model to maintain the image context represented as the transferred brain signal but also benefits the later natural fusion of brain context. The fine-tuning of the diffusion model with the parameters $\theta$ can be formulated as:

$$\mathcal{L}_t^{\text{img}}(\theta) = \mathbb{E}_{x,\epsilon\sim\mathcal{N}(0,1),t}\left[\|\epsilon - \epsilon_\theta(x_t, t, e_p)\|_2^2\right], \tag{6}$$

where $t = 1, \ldots, T$ and $x_t$ is Gaussian corrupted image of input $I$. $\epsilon_\theta(x_t, t)$ is a UNets-based denoising function.

### Context Input

Image    Brain

### Image context gradually interpolated with brain context

α = 0.2    0.4    0.5    0.6    0.8    1

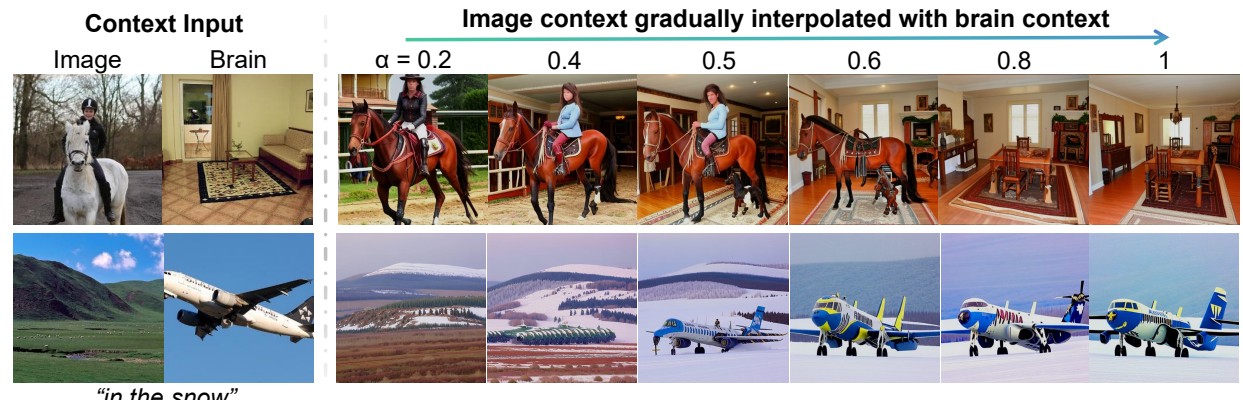

*"in the snow"*

*Figure 5.* Visualization of the generation process through gradual interpolation between image and brain contexts.

**Img / Brain**

**Img / Brain**

**Img / Brain**

**Img / Brain**

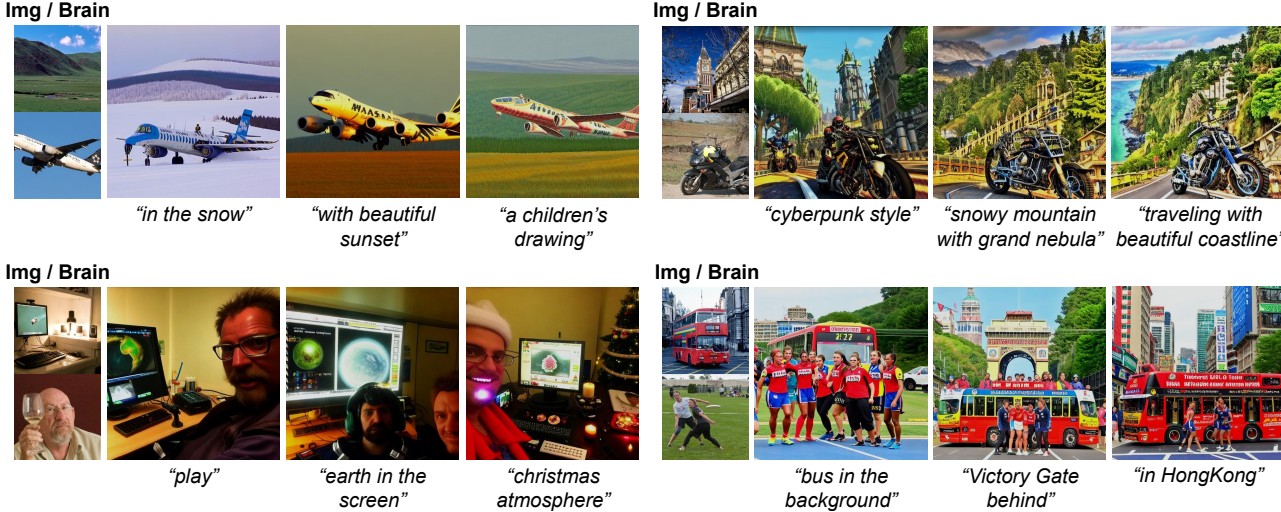

*"in the snow"*    *"with beautiful sunset"*    *"a children's drawing"*

*"cyberpunk style"*    *"snowy mountain with grand nebula"*    *"traveling with beautiful coastline"*

*"play"*    *"earth in the screen"*    *"christmas atmosphere"*

*"bus in the background"*    *"Victory Gate behind"*    *"in HongKong"*

*Figure 6.* Multi-context blending results with only different text contexts. MindCustomer robustly creates multi-context images that are content-consistent and naturally integrated.

Next, we empirically observe that when different contexts are fused often results in semantic conflict. Partial context may lost or disturbed. To better preserve the multi-semantics and enhance seamless fusion, we lightly optimize the brain context embedding $e_b$ for approximating the semantics of the image context. The optimization only modifies the brain context embedding while we freeze the parameters of the diffusion model. We run this optimization for a few steps and this process can be formulated as:

$$\mathcal{L}_t^{\text{brain}}(e_b) = \mathbb{E}_{x,\epsilon \sim \mathcal{N}(0,1),t} \left[ \| \epsilon - \epsilon_\theta(x_t, t, e_b) \|_2^2 \right], \quad (7)$$

Consequently, we can adapt the semantics between the image and brain contexts in latent space that enables the diffusion model to better understand the fusion of both embeddings to generate natural images.

**Embeddings Integration** Finally, we apply the fine-tuned

diffusion for blended image outcomes. As for the generation only guided by dual contexts, we directly concatenate the embeddings of $e_p$ and $e_b$ for the VD to obtain the results. As for triplet contexts, the concatenated dimension of embeddings will exceed the model capacity. Thus, we utilize linear interpolation of $e_p$ and optimized $e_b$ with a hyperparameter $\alpha \in [0, 1]$ to build the combined embedding $e_c$:

$$e_c = (1 - \alpha) \cdot e_p + \alpha \cdot e_b. \quad (8)$$

We employ the combined embedding $e_c$ to the image flow of the fine-tuned versatile diffusion, and meanwhile feed the text context embedding to the text flow, to jointly generate the final multi-context blended image. Note that the $\alpha$ can be smoothly adjusted for asymptotical generation results balancing the image and brain contexts.

**New-subject Few-shot Generation** Collecting brain signals in real-world scenarios is time-consuming and costly, often resulting in limited data for new subjects. To address

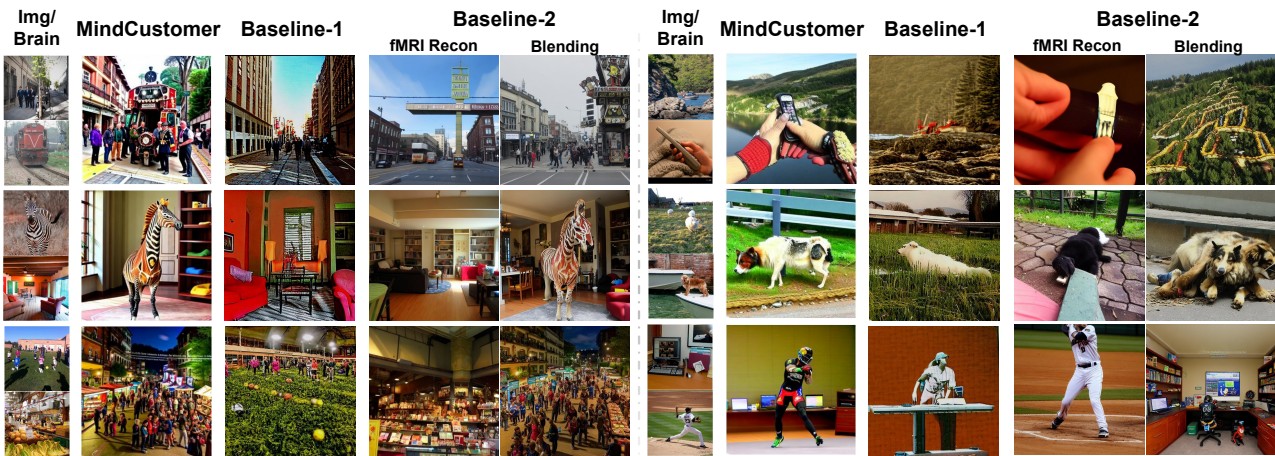

*Figure 7.* Baseline comparison. Baseline-1: We replace the brain context with its visual stimuli to construct a multi-image blending. Baseline-2: We reconstruct the brain context to the image and merge all the image contexts for blending. Benefiting from the designed cross-modal fusion pipeline, MindCustomer naturally preserves and fuses different contexts to generate more visually pleasing results.

*Table 1.* Quantitative comparisons. We compare our method with the baselines on context similarity and generation quality.

|            | Baseline-1 | Baseline-2 | Ours      |
|------------|------------|------------|-----------|
| CLIP-I↑    | 0.559      | 0.526      | **0.563** |
| DINOv2↑    | 0.630      | 0.612      | **0.654** |
| CLIP-IQA↑  | 0.734      | 0.830      | **0.892** |

this, MindCustomer constructs a cross-subject model that integrates brain signals from different subjects into a unified framework, enabling better generalization to new subjects. Specifically, we first use limited new-subject data to train a new IBT and fine-tune the brain embedding network. Leveraging the Reset-Tuning operation (Wang et al., 2024), we train only the shallow embedder for the new subject while freezing the deep shared embedder, without IBT augmentation. The remaining generation steps are the same as the above processes. Empirical results show that using less than 20% of the full dataset is sufficient for few-shot learning to generate natural outputs.

## 4. Experiments

**Evaluation Metrics** To assess the semantic preservation of the generated image, we use the common metrics on image similarity, DINOv2 (Oquab et al., 2024) and CLIP-I (Radford et al., 2021c). We further apply CLIP-IQA (Wang et al., 2023) to test the generation quality, a measure evaluating both the visual appeal and the conceptual perception of an image. Moreover, we adopt metrics of visual fMRI reconstruction to assess brain decoding performance. Following previous works (Wang et al., 2024; Xia et al., 2024), Pix-Corr, SSIM (Wang et al., 2004), AlexNet(2), and AlexNet(5)

(Krizhevsky et al., 2012) are low-level property evaluation, and metrics of Inception (Szegedy et al., 2016) and CLIP (Radford et al., 2021a) are high-level property evaluation. The implementation is described in **Appendix-B**.

### 4.1. Qualitative Results

We use MindCustomer for multi-context image generation by integrating semantic information from brain signals. Image contexts from the NSD COCO dataset are paired with brain contexts and optionally enhanced with concise textual prompts to control spatial, environmental, and stylistic attributes. We present the customization results across all the subjects of NSD in Figure 3, showcasing the generation of high-quality and contextually faithful images. More results are presented in **Appendix-D.1**.

Since the visual stimuli in test data are shared across all four subjects, we conduct experiments under identical conditions using the same contexts for each subject. This setup allows us to evaluate the model's robustness and showcase the individual variability among different subjects. The results are shown in Figure 4.

In the embedding interpolation of MindCustomer, we apply a hyperparameter $\alpha$ to control the ratio of transferred image embedding and brain instruction embedding. In Figure 5, we visualize the generation process via smoothly changing $\alpha$. It can be observed that when the alpha value is around 0.5, our method achieves a natural fusion of two embeddings while preserving their respective feature information. Empirical experiments indicate that $\alpha$ range between 0.4 and 0.7 generally yields optimal blending results.

Moreover, we present more visual results in Figure 6 with only different text contexts. As we can see, MindCus-

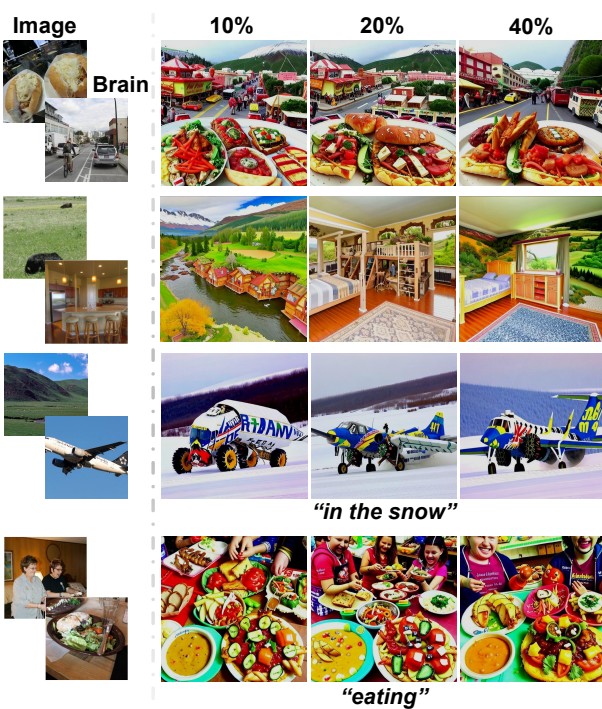

**Figure 8.** Few-shot generation to the new human subject. We separately utilize 10%, 20%, and 40% of the subject's complete dataset to perform generation fine-tuning. MindCustomer generalizes well to the new subject, indicating its real-world applicability.

tomer robustly creates multi-context images that are content-consistent and naturally integrated.

## 4.2. Comparisons and Analysis

**Baseline** Unlike traditional methods (Ruiz et al., 2023; Ding et al., 2024; Kumari et al., 2023), relying on masks or multiple images of the same object, MindCustomer enables mask-free, single-image generation, which makes direct comparisons challenging. To address this, we design two baselines using versatile diffusion (VD) (Xu et al., 2023), capable of multi-context blending. **Baseline-1**: We replace the brain context with its image-based visual stimuli and input all the contexts for VD to generate results. **Baseline-2**: We use an SOTA fMRI-to-image model (MindEye (Scotti et al., 2023)) to reconstruct the brain context as another image context, and merge all the contexts for VD blending. Note that as baselines, VD requires no extra fine-tuning or optimization and relies on the original large text-image pre-training capability for direct inference.

Here we present both qualitative and quantitative comparisons of generated images in Figure 7 and Table 1. Due to the *brain signal interpretation error* caused by fMRI reconstruction and *some semantic overlap* between distinct image contexts resulting from the blending diffusion model, these baselines show some semantic deviation and loss. While

**Table 2.** User study on multi-context image generation. Users are asked to score the generated images from 1 to 3 (3 is the best).

| Method | Quality↑ | Consistency↑ |
|---|---|---|
| Baseline | 1.868 | 1.675 |
| Ours | 2.618 | 1.868 |

**Table 3.** Quantitative results on Subj1 few-shot generation.

| | 10% | 20% | 40% |
|---|---|---|---|
| CLIP-I↑ | 0.523 | 0.541 | 0.561 |
| DINOv2↑ | 0.642 | 0.650 | 0.652 |
| CLIP-IQA↑ | 0.944 | 0.947 | 0.952 |
| Average ↑ | 0.703 | 0.713 | 0.722 |

in MindCustomer, we designed VD fine-tuning based on image context to learn the image semantics, and then we freeze the fine-tuned VD model, and lightly optimize brain contexts with the target of the image context. This helps mitigate semantic conflicts between different contexts. During the above process, we further utilize the IBT to transfer image context to brain-embedding space to reduce the gap between modalities. Thus, *the designed brain-blended cross-modal fusion pipeline* allows the blending to better preserve each modality's semantics and produce more accurate, high-quality results. The quantitative metrics also demonstrate that MindCustomer significantly outperforms the baselines in terms of context similarity and image generation quality.

**User Study** We also conduct the human evaluation study, surveying our method and baseline. As the overall performance of Baseline-1 surpasses that of Baseline-2, we choose Baseline-1 as the baseline for this analysis. In the study, each generated image is scored according to the generation quality and the alignment to the contexts. In total, we collect 220 answers, whose results are summarized in Table 2. Participants exhibit a notable preference for our method in both perspectives. More details of the study protocols are described in **Appendix-C**.

## 4.3. Few-shot Generation on New Subject

In real-world scenarios, collecting brain signals from human subjects is highly costly, leading to data scarcity. Therefore, the ability of a designed algorithm to generalize to new individuals is crucial. To demonstrate MindCustomer's practical generalization capability, we pre-trained the model from Subj2, 5, and 7 and performed few-shot learning on Subj1. We fine-tuned our model using 10%, 20%, and 40% of the total data from Subj1, respectively.

As shown in Figure 8, MindCustomer achieved natural and competitive generation results even with limited data for fine-tuning. As the data point increased, the model consistently generated higher-quality images. The quantitative

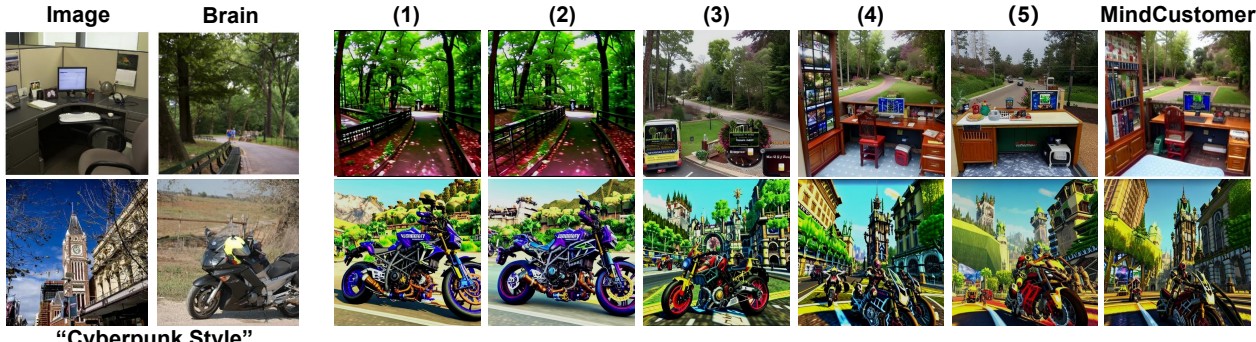

*Figure 9.* Qualitative ablation. **(1) wo [IBT & fine-tuning & optimization]**: Direct generation by concatenating CLIP-encoded image and Brain embedder-encoded contexts leads to semantic inconsistency and context overlap. **(2) wo IBT**: Adding fine-tuning and optimization fails to resolve these issues. **(3) wo [fine-tuning & optimization]**: Using IBT for alignment before concatenation allows partial explicit representation of semantic information. **(4) wo optimization**: Adding fine-tuning significantly improves results, enabling better representation of different semantic contexts. **(5) wo ClipCap**: Without the introduced ClipCap for semantic enhancement in Brain Representation Pretraining may result in low details of context representation in blending results. MindCustomer: Incorporating all techniques, we can produce higher-quality image generation with detailed context fidelity.

*Table 4.* Quantitative ablation. We compare MindCustomer with the ablations on context similarity and generation quality. (1) wo [IBT & fine-tuning & optimization]; (2) wo IBT; (3) wo [fine-tuning & optimization]; (4) wo optimization; (5) wo ClipCap.

|  | (1) | (2) | (3) | (4) | (5) | **MindCustomer** |
|---|---|---|---|---|---|---|
| CLIP-I↑ | 0.518 | 0.515 | 0.529 | 0.548 | 0.560 | **0.563** |
| DINOv2↑ | 0.594 | 0.595 | 0.614 | 0.635 | 0.637 | **0.654** |
| CLIP-IQA↑ | 0.842 | 0.838 | 0.893 | 0.893 | 0.875 | **0.892** |

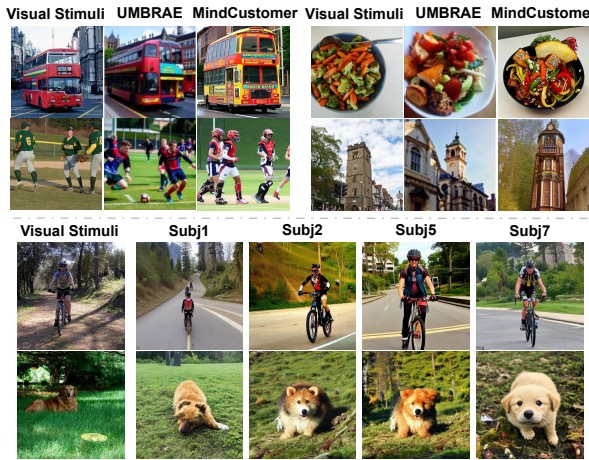

*Figure 10.* Above: Comparison of brain decoding on Subj1. Below: Brain decoding across subjects by MindCustomer.

results of few-shot learning under different data volumes are presented in Table 3, indicating the promising generalization ability of MindCustomer. More few-shot generation results are illustrated in **Appendix-D.1**.

### 4.4. Ablation

We conduct experiments to validate the components of MindCustomer. First, we design IBT to achieve cross-modal alignment by mapping semantic representations from different modalities into a shared implicit space, forming the basis for fusion. Second, we fine-tune the diffusion model on image contexts to preserve semantics and optimize brain contexts for high-level semantic adaptation, generating coherent images. Figure 9 illustrates the key components of our design, and Table 4 provides quantitative ablation results. Besides, we visualize the brain reconstruction ability of MindCustomer in Figure 10. Compared with the SOTA method (Xia et al., 2024), MindCustomer also demonstrates superiority and robustness across diverse subjects. The quantitative results on brain reconstruction and more detailed ablation experiments are conducted in **Appendix-D**.

## 5. Conclusion

In this paper, we propose a novel brain-blended image customization method, MindCustomer. We introduce the IBT to enhance the data augmentation for cross-subject brain embedding. Further, the proposed fusion information pipeline, along with IBT, effectively adapts distinct semantics across multiple contexts. Thus, MindCustomer can generate context-preserved and naturally coherent blending results. Besides, it can also generalize to new human subjects via few-shot learning, demonstrating the potential for real-world applications. In all, we hope MindCustomer serves as an exploration of brain context in the image customization task and gives inspiration for follow-up research, promoting the development of brain-computer interaction.

## Acknowledgements

This work was supported by the Beijing Natural Science Foundation (Grant No.JQ24023 and No.F251020), Beijing Municipal Science & Technology Commission Project (No.Z231100006623010), National Science and Technology Major Project (2022ZD01163013 to B.L.), and Beijing Municipal Science and Technology Project (No.Z241100004224028).

## Impact Statement

With the development of generative models and large language models, AI-driven personalized generation is garnering increasing attention and application. At the same time, advancements in neural data decoding are empowering researchers to delve deeper into brain-machine interfaces (BMIs) and human-computer interaction (HCI). The integration of BMIs and generative models is poised to make personalized brain-controlled generation a significant trend in the future of generative artificial intelligence. Mind-Customer represents the first exploration of multi-context image generation guided by brain semantics. We aim for it to serve as foundational research, offering potential insights and inspiration for future research in related fields as well as practical applications. We believe personalized brain-controlled generation will lead to groundbreaking developments in areas such as artistic creation, game design, and advertising industries.

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

# Appendix

***Note that all the appendix figures are at the end of the text.***

## A. More Related Works

Apart from the image customization, blending and composition introduced in the main paper, here we describe other related works on editing task. Image editing applies more precise control to the source image, allowing users to slightly modify image contents according to external instructions. Many works perform image manipulation based on GAN models (Abdal et al., 2021; Härkönen et al., 2020; Lang et al., 2021; Patashnik et al., 2021; Shen et al., 2020; Shen & Zhou, 2021) to edit images with contents and styles. Recently, the success of the diffusion model (Rombach et al., 2022; Xu et al., 2023) and large-scale pre-trained CLIP model (Radford et al., 2021b) also improves the development of editing (Avrahami et al., 2022; Hertz et al., 2023b; Ramesh et al., 2022b). Several methods integrate text conditions into the denoising process of latent diffusion to alter the resulting images (Brooks et al., 2023; Kawar et al., 2023). Liu et al. (Liu et al., 2023a) trained diffusion models to generate images that approximate the text condition, enabling edits to the source images. Hertz et al. (Hertz et al., 2023a) enhanced cross-attention mechanisms to more accurately fuse the source image with the condition, allowing for more precise modifications. Concurrently, research on image inpainting (Yang et al., 2023; Wasserman et al., 2024) has focused on utilizing manually added masks to incorporate text conditions, facilitating targeted image editing. In all, image editing focuses on more specific modifications to the reference image while customization and blending mainly deal with context integration.

## B. Implementation

We utilize the training set of NSD data for the Brain Representation Pre-training, and use the test set as the brain context for generation. We train the subject-wise IBT with a learning rate of $5e^{-5}$ for 200 epochs and then adopt AdamW schedule for our Brain Embedder with a learning rate of $3e^{-3}$ for 600 epochs. MindCustomer is implemented with the publicly available versatile diffusion. The model supports multi-modal generation with multiple flows in the latent space (of size 4 x 64 x 64), accompanied by VAE encoders and decoders for image and text. We fine-tune the VD for 200 epochs with a learning rate of $5e^{-8}$ utilizing Adam schedule. Then we optimize the brain embedding for 100 epochs with a learning rate of $1e^{-5}$. We present the intuitively detailed setting in Table 5. The time of one image generation is about 6 minutes on a single Tesla A100 GPU. The efficiency enables us to perform single-image real-time creation for MindCustomer.

*Table 5.* Training parameters of MindCustomer

|  | Optimizer | Learning rate | Weight decay | Training epochs | Batch size | LR scheduler |
|---|---|---|---|---|---|---|
| IBT | AdamW | $5e^{-5}$ | $1e^{-2}$ | 200 | 50 | OneCycleLR |
| Brain Representation | AdamW | $3e^{-3}$ | $1e^{-2}$ | 600 | 50 | OneCycleLR |
| VD Finetune | Adam | $5e^{-8}$ | 0 | 200 | 1 | None |
| Embedding Opt | Adam | $1e^{-5}$ | 0 | 100 | 1 | None |

## C. Details of User Study

We randomly selected 110 pairs of comparison images, including ours and the baseline, and divided them into 11 groups. A total of 22 participants were involved, and to minimize bias caused by individual preferences, we randomly assigned every two participants to the same group, ensuring that each image received ratings from two different individuals. By comparing ours with the baseline, participants are required to score each generated image based on the quality and consistency from 1 to 3, with 3 being better. Ultimately, we collected 220 answers for the 110 randomly selected image pairs from the 22 participants. The results broadly reflect the participants' consistent preference for our approach.

*Table 6.* Pearson correlation coefficient and significance analysis on brain simulation of IBT with the ground truth. The range of Pearson and significance coefficient $\alpha$ is [-1, 1] and [0, 1].

|  | **Subj1** | **Subj2** | **Subj5** | **Subj7** | **Avg** |
|---|---|---|---|---|---|
| **Pearson** | 0.389 | 0.423 | 0.502 | 0.389 | 0.426 |
| $\alpha$ | $1 - 5.73 \times 10^{-8}$ | $1 - 1.170 \times 10^{-12}$ | $1 - 7.370 \times 10^{-19}$ | $1 - 9.250 \times 10^{-4}$ | 0.999 |

# D. More Experiments

## D.1. Additional Results on Multi-Context Generation

In Figure 11, we provide more results of multi-context generation across different subjects. And in Figure 12, we illustrate more few-shot generation results on Subj1.

## D.2. Different Seeds for Generation

To demonstrate the robustness of MindCustomer, we present generation results across different random seeds in Figure 13. Our method is capable of producing images of consistently high quality, with minor variations. We also recommend that users experiment with multiple random seeds to obtain results that best suit their preferences.

## D.3. Discussion on Embedding Integration

In this paper, we employ two different methods of embedding fusion: interpolation and concatenation. In the triplet contexts generation, we apply linear interpolation to the image and brain contexts to prevent exceeding the dimensional capacity of the VD model. In the case of dual contexts, we directly concatenate the two. To demonstrate the impact of these two fusion methods on generation quality, we input identical image and brain contexts with both fusion strategies in the absence of text context. In Figure 14, both methods successfully merge the information in a natural manner. However, we observe that, compared to concatenation, linear interpolation slightly alters the original semantic representations of the two modalities in the latent space, resulting in generated images with somewhat reduced fidelity and details. We speculate that this issue may be addressed through more sophisticated mechanisms, such as cross-attention. Additionally, optimizing this process with the currently limited brain data presents a significant challenge. Nevertheless, our method shows that even with simple fusion techniques, satisfactory results can be achieved. As research on neural data continues to evolve, we anticipate the development of more effective mechanisms to generate even higher-quality images.

## D.4. Text Context for Enhanced Generation

In Figure 15, we compare the impact of including versus excluding the text context on generation results using the same image and brain contexts. It is evident that the text context provides additional semantic information or more precise interaction control between the image and brain contexts. This empowers MindCustomer to better meet user needs, enabling more accurate, richer, and higher-quality image generation.

## D.5. Effectiveness of Image-Brain-Translator

We conduct experiments to demonstrate the effectiveness of IBT in simulating pseudo-brain signals. First, we simulate brain signals for visual stimulus images from the test set and quantitatively assess them against ground truths (GT). Following standard practices in neuroscience for brain signal simulation, we measure the similarity between the simulated and GT brain signals using the *Pearson correlation coefficient*. In Table 6, the average Pearson correlation coefficient across the four subjects is 0.426 (ranging from -1 to 1). According to relevant literature (Chee, 2015; Evans, 1995), it indicates a moderate positive correlation, meanwhile with a high confidence level of 99%. The results show that IBT can approximate the brain signal waves corresponding to the input images, providing a foundational prior for natural cross-modal fusion. Additionally, we visualize the brain signal simulations for four human subjects in Figure 16, further validating the method's effectiveness. Moreover, we transfer the brain signals derived from the images and directly fed them in the diffusion model to explicitly showcase the semantic accuracy of the simulated brain signals in Figure 17.

*Table 7.* Ablations on brain decoding. We compare our method with the open-sourced SOTA methods. For a fair comparison, all methods are trained on four subjects from NSD with only one generative model.

| Method | PixCorr↑ | SSIM↑ | AlexNet(2)↑ | AlexNet(5)↑ | Inception↑ | CLIP↑ |
|---|---|---|---|---|---|---|
| MindBridge (Wang et al., 2024) | .151 | .263 | 87.7% | 95.5% | 92.4% | 94.7% |
| UMBRAE (Xia et al., 2024) | .283 | .328 | 93.9% | 96.7% | 91.7% | 93.5% |
| Ours | .157 | .289 | 88.3% | 95.5% | 92.1% | 94.5% |
| wo IBT | .152 | .272 | 87.9% | 95.2% | 91.5% | 94.1% |

## D.6. Additional Results on Brain Decoding

Table 7 shows the reconstruction performance, which highlights the impact of our Brain Representation Pre-training, where reconstructing brain signals with IBT improves cross-subject embeddings and achieves competitive results with SOTA methods (Wang et al., 2024; Xia et al., 2024). In Figure 18 & 19 & 20, we conduct more brain reconstruction comparisons with another SOTA MindBridge (Wang et al., 2024) and illustrations to demonstrate the effectiveness of our Brain Representation Pre-training. Moreover, we further illustrate the few-shot reconstruction capability of MindCustomer, as shown in Figure 20, where our method can also precisely reconstruct the visual stimuli under limited data.

## E. Discussion and Limitation

MindCustomer utilizes visual brain signals as one of the key information sources in multi-context image generation. Here we discuss our approach with traditional methods (Alaluf et al., 2023; Cao et al., 2023; Chen et al., 2024b; Gal et al., 2023b; Tewel et al., 2023b) that rely solely on image and text modalities in three aspects. (1) Data: Brain signal acquisition directly and implicitly reflects the user's intentions, without the need for additional modality conversion. In contrast, traditional methods often require users to explicitly convey their ideas through images or text, which may introduce deviations and complexities. (2) Technology: Traditional methods can directly embed prompts with the assistance of large-scale pre-trained models, such as CLIP and diffusion models. In contrast, brain-controlled image customization presents more challenges, requiring precise embedding of brain signals and bridging the implicit gaps between modalities to achieve naturally blended results. (3) Result: Since text and images can provide fine-grained semantic information, image generation (e.g. customization, editing, manipulation) based on these modalities can achieve more detailed alignment. However, current brain signals primarily record coarse information due to noise or disturbance, making them more suitable for global context blending but potentially insufficient for handling detailed features or specific instructions.

Besides, prior works that focus on better decoding brain signals themselves (Scotti et al., 2023; 2024; Wang et al., 2024; Xia et al., 2024) and utilize neural data for image stylization (Chen et al., 2024a) or painting (Yu et al., 2025) have made tremendous progress in brain analysis and applications. Different from these works, MindCustomer further researches how to fuse brain signals with other contexts in image and text modalities to mask-freely create naturally blended images. Our experimental results have proven that MindCustomer is capable of effectively extracting brain signal semantics, and more importantly, it has demonstrated excellent performance in the personalized blending task. Therefore, we believe that previous works and MindCustomer can mutually inspire and complement each other, sparking more interesting and meaningful research and applications in the field of brain-computer interaction.

The limitation of this work primarily lies in current brain signal acquisition and analysis, which requires further development. This includes improving the quality (e.g. Signal to Noise Ratio) and diversity (e.g. type, content) of neural data, and expanding data samples, as these factors currently constrain our precision and richness of personalized generation. Further, we believe that the collection of brain imagery data specifically for brain-controlled generation tasks may inject new momentum into the development of this field. In the future, we will also explore and acquire these data. As brain-machine interface technology continues to evolve, brain-controlled creative research will become increasingly sophisticated. We hope MindCustomer provides a critical foundation for advancing this emerging area of research.

| Image | Brain | MindCustomer | Image | Brain | MindCustomer |
|---|---|---|---|---|---|

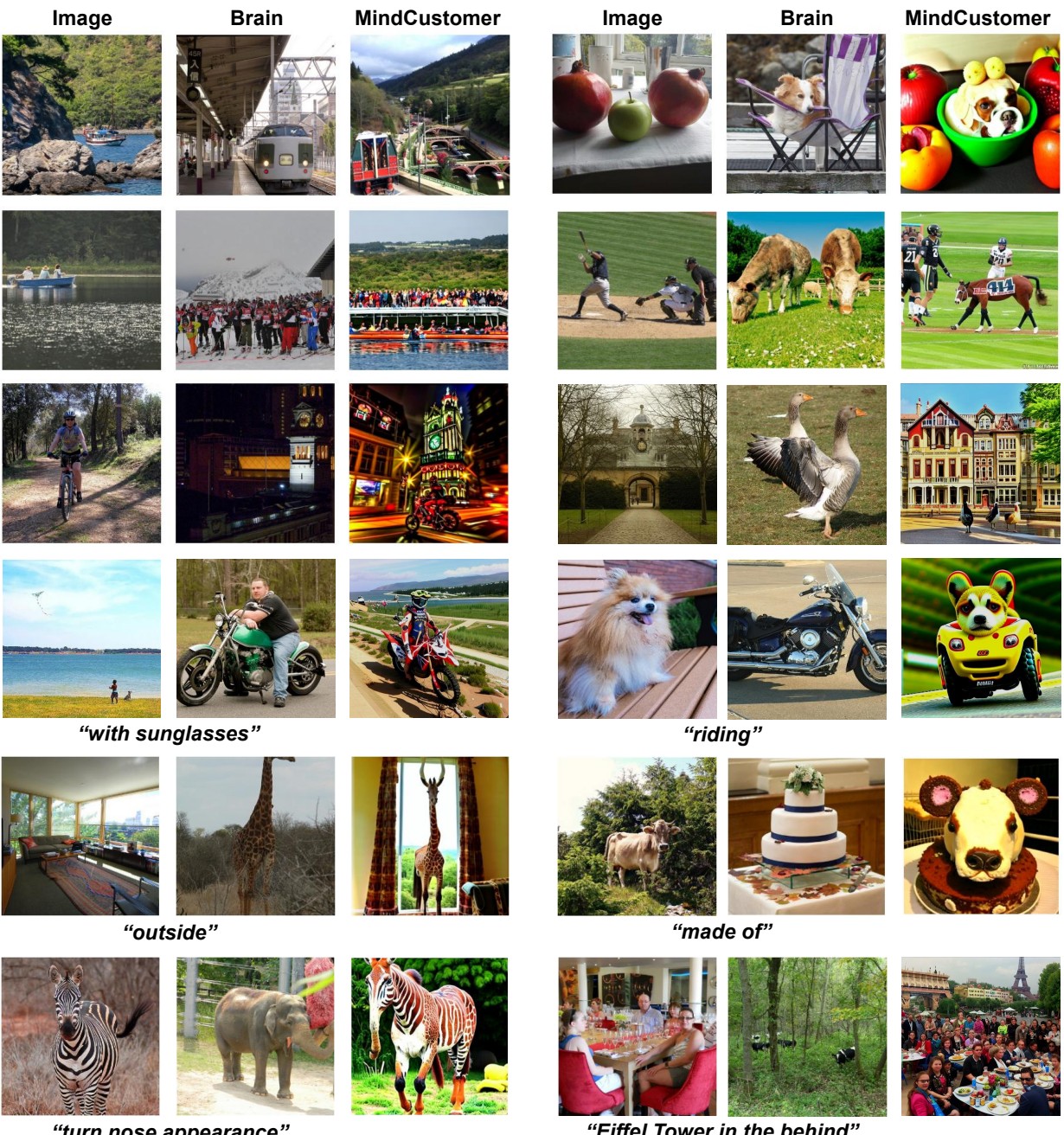

*"with sunglasses"*        *"riding"*

*"outside"*        *"made of"*

*"turn nose appearance"*        *"Eiffel Tower in the behind"*

*Figure 11.* More brain-blended multi-context generation results.

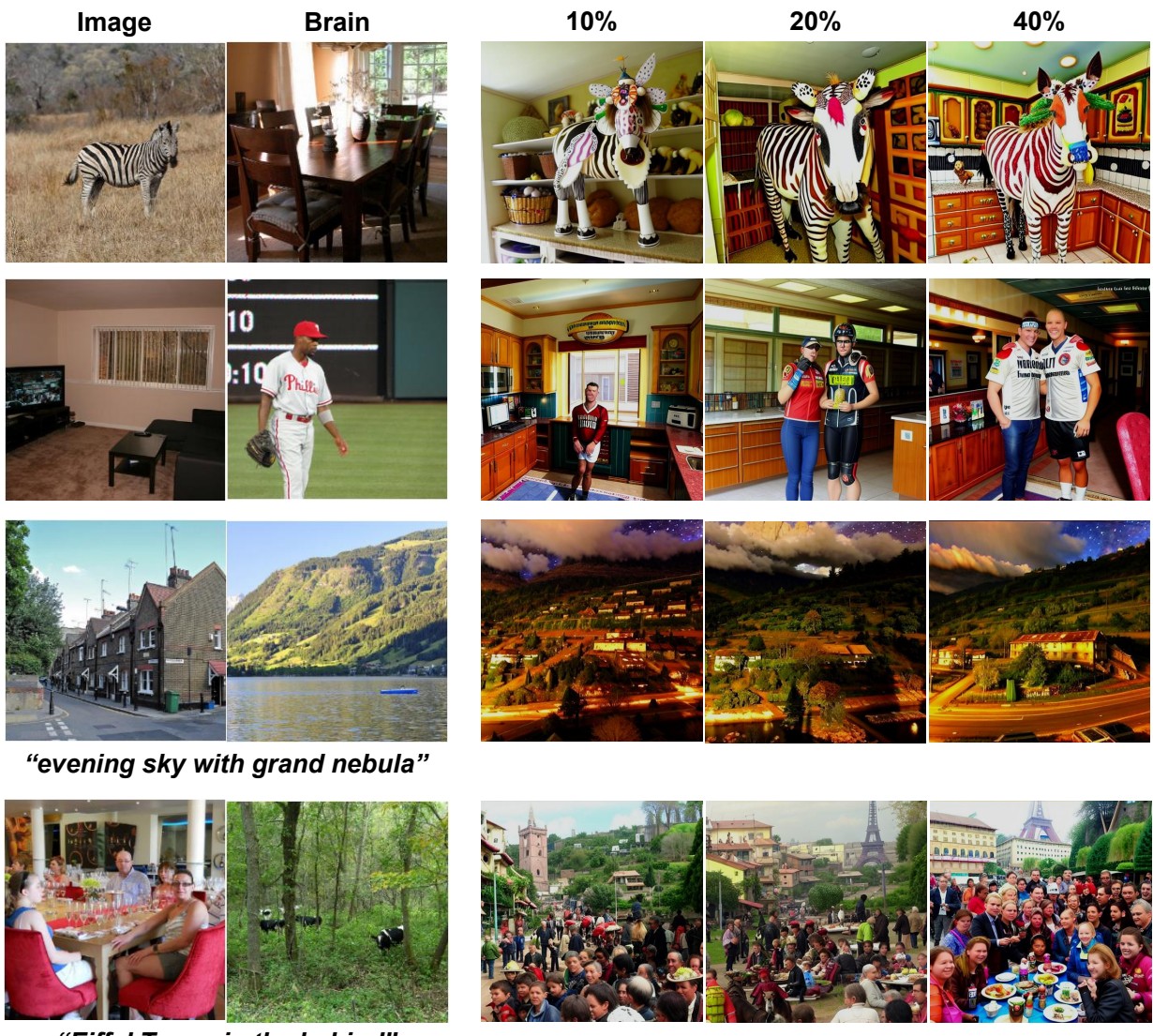

*Figure 12.* More few-shot generation results on Subj1. We only use 10%, 20%, and 40% of the subject's complete training data for few-shot learning.

**Image**  **Brain**  **MindCustomer**

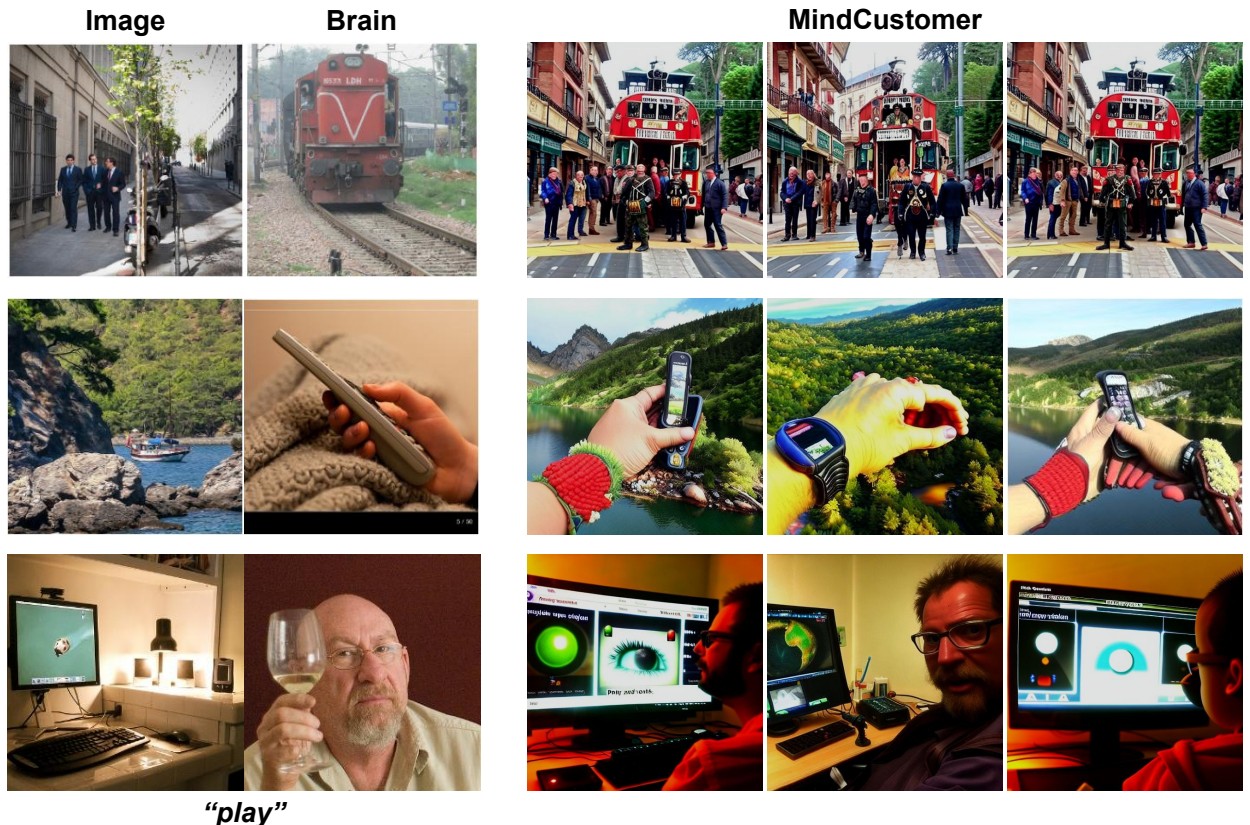

*"play"*

*Figure 13.* Different seeds for generation.

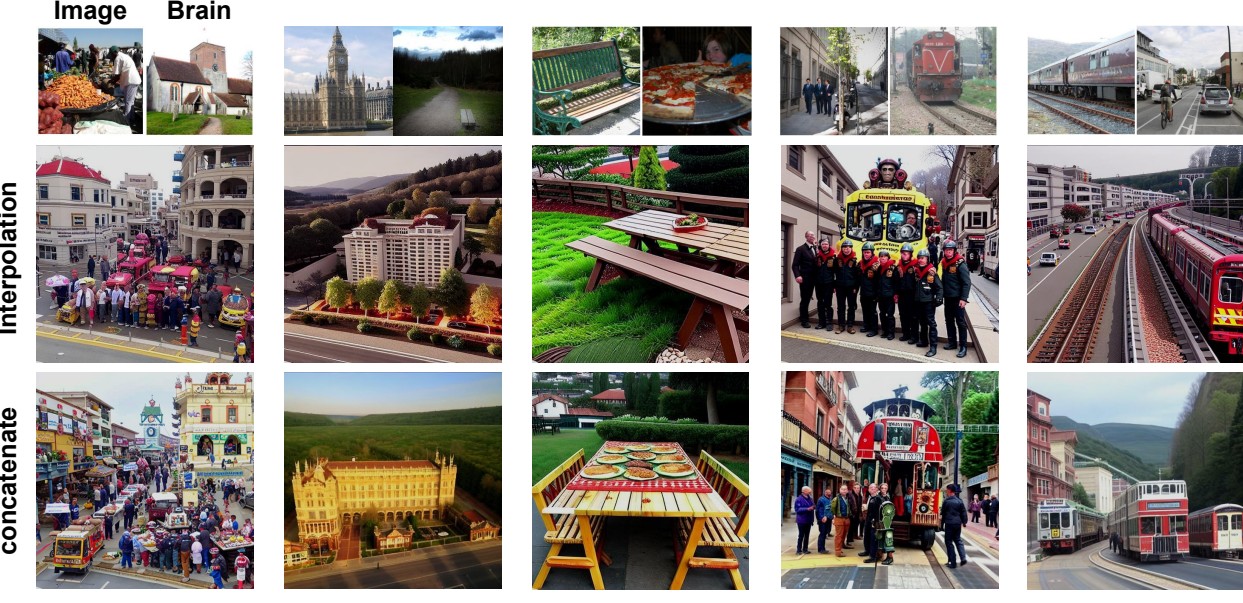

*Figure 14.* Comparison on the methods of brain and image embedding integration: interpolation and concatenation.

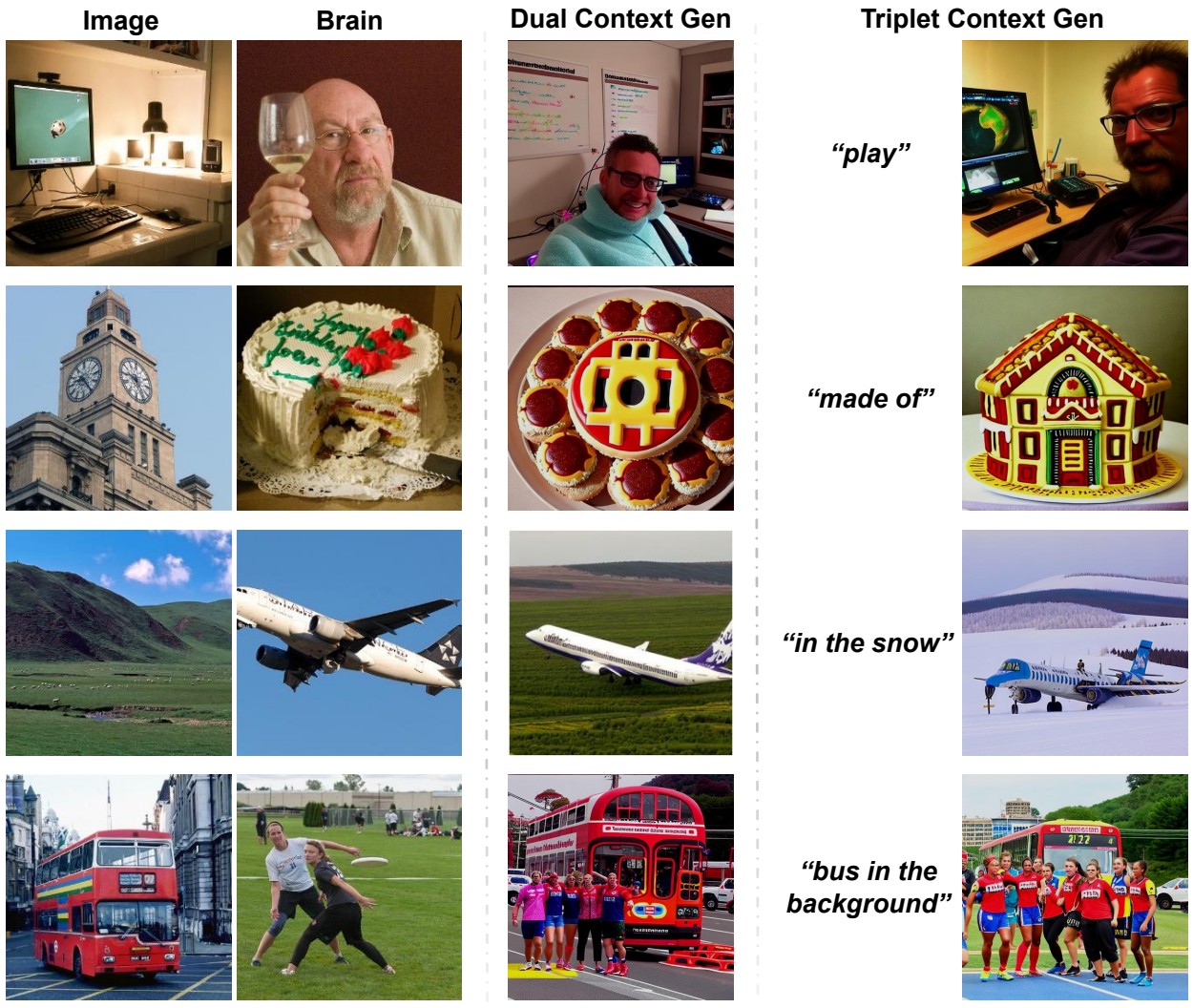

Figure 15. Generation results w / wo text contexts. It demonstrates that the text context provides additional semantic information or more precise control over the image and brain contexts.

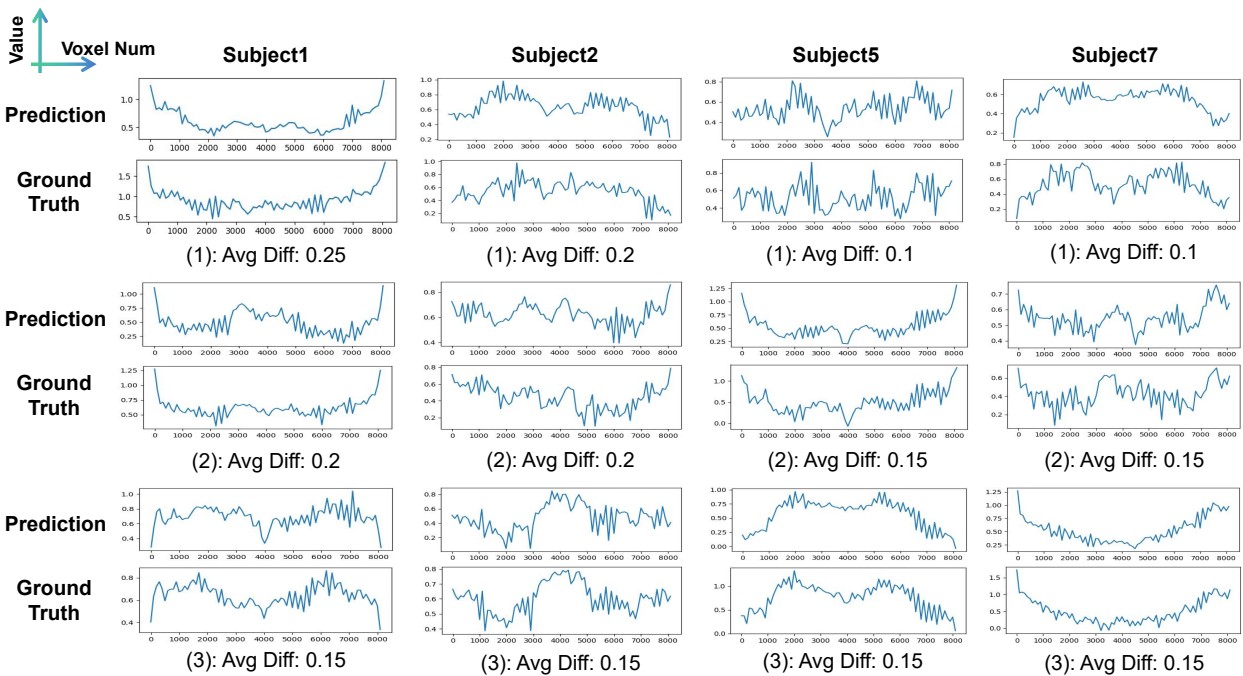

*Figure 16.* Visualization of the predicted brain voxel by IBT with its ground truth (GT). It illustrates voxels in brain signals with their corresponding feature values. Here we randomly choose three examples for each subject, and report the difference error between the prediction and the ground truth. We simply compute the absolute difference in value for each voxel, and then take the average of the absolute differences across all voxels, symbolized as Avg Diff. Note that the range of Avg Diff is [0, 2], lower is better. It is observed that the subject-wise IBT can effectively simulate brain signals with low differences to GT. It facilitates the transferring accuracy of image features to brain modality in our method.

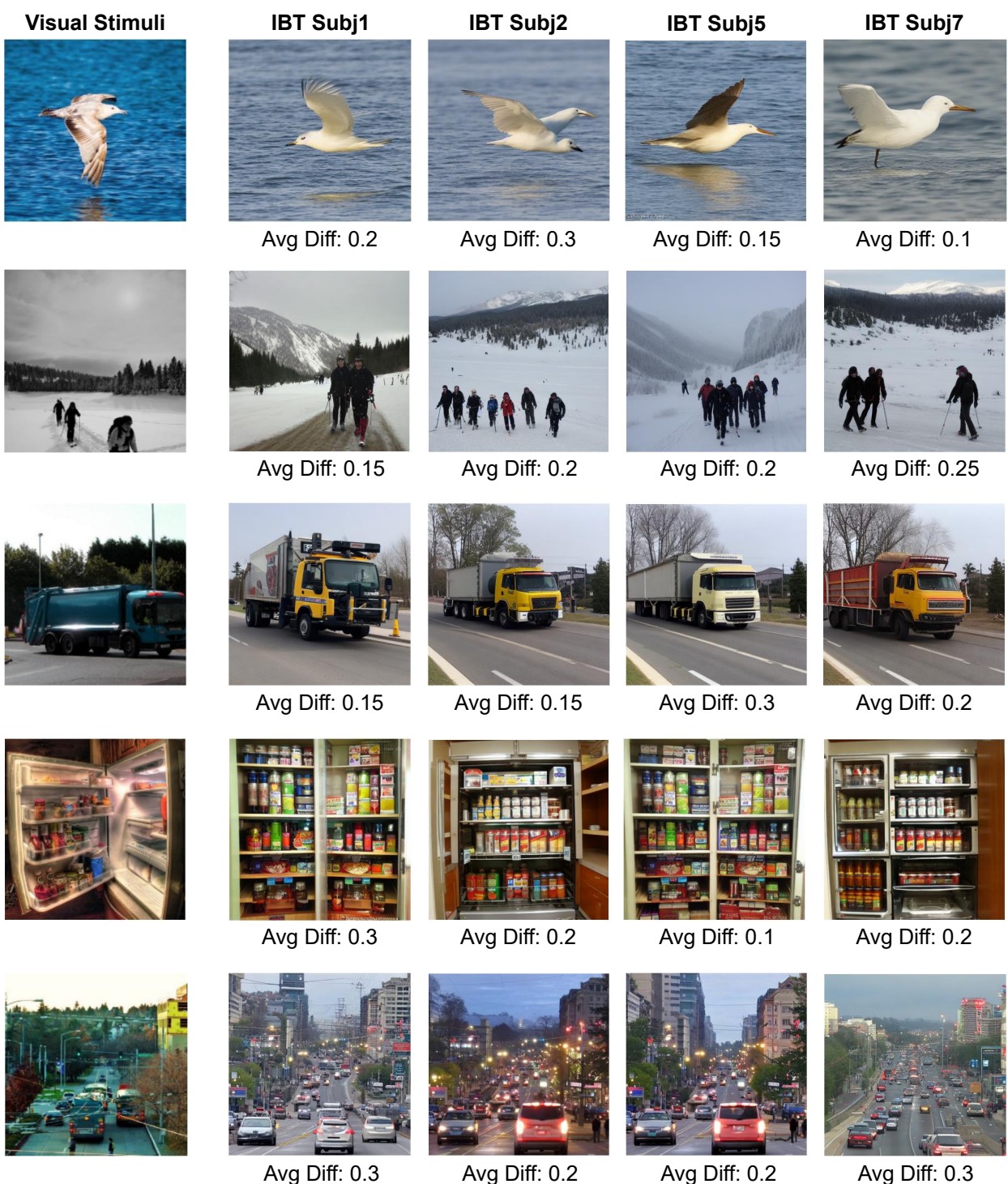

*Figure 17.* We utilize the predicted brain voxels of IBT for direct reconstruction to explicitly demonstrate the effectiveness of accurate semantics. Same with Figure 16, here we also report the Avg Diff of brain voxels between the prediction and the ground truth.

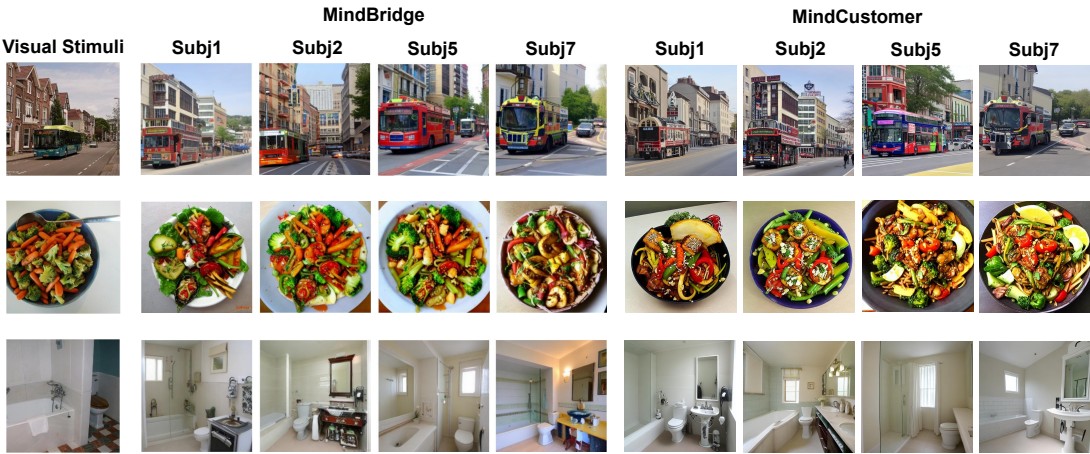

*Figure 18.* Comparison on the cross-subject brain reconstruction. We compare our method with the SOTA MindBridge (Wang et al., 2024) to show its competitive performance of MindCustomer on the brain decoding task.

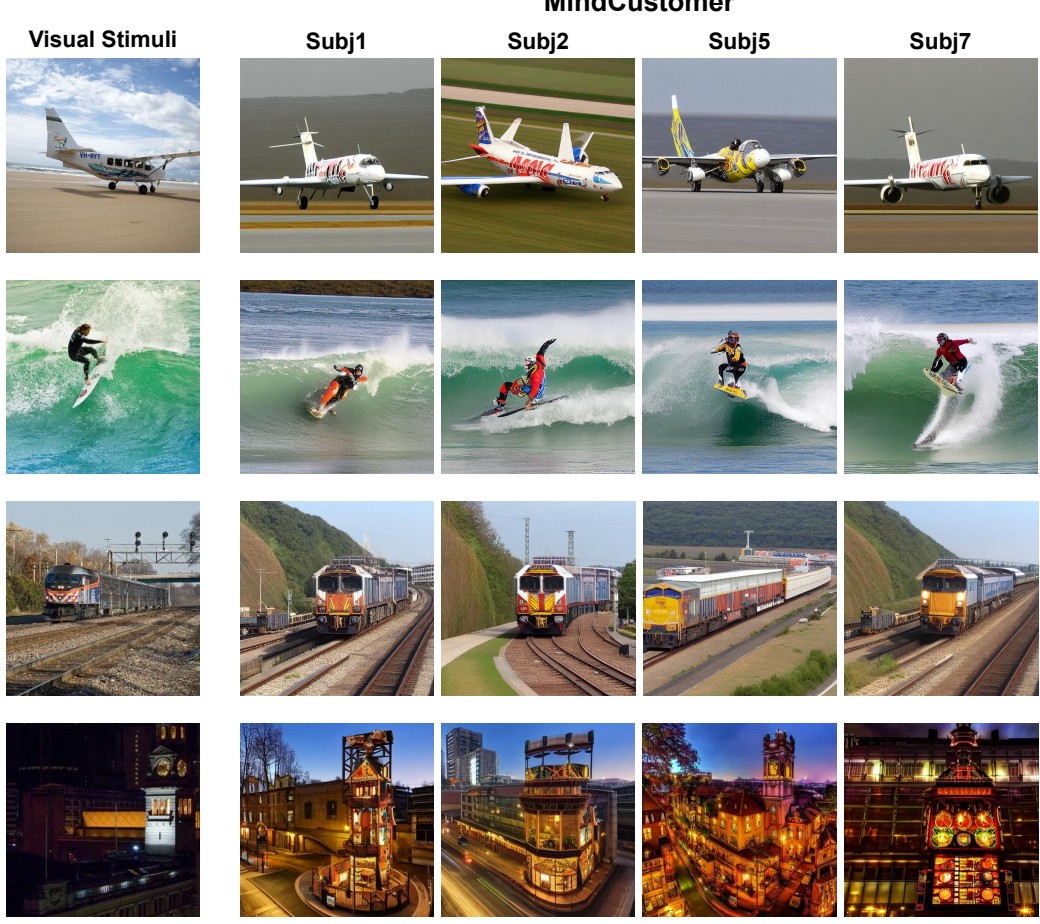

*Figure 19.* More cross-subject brain reconstruction results of MindCustomer.

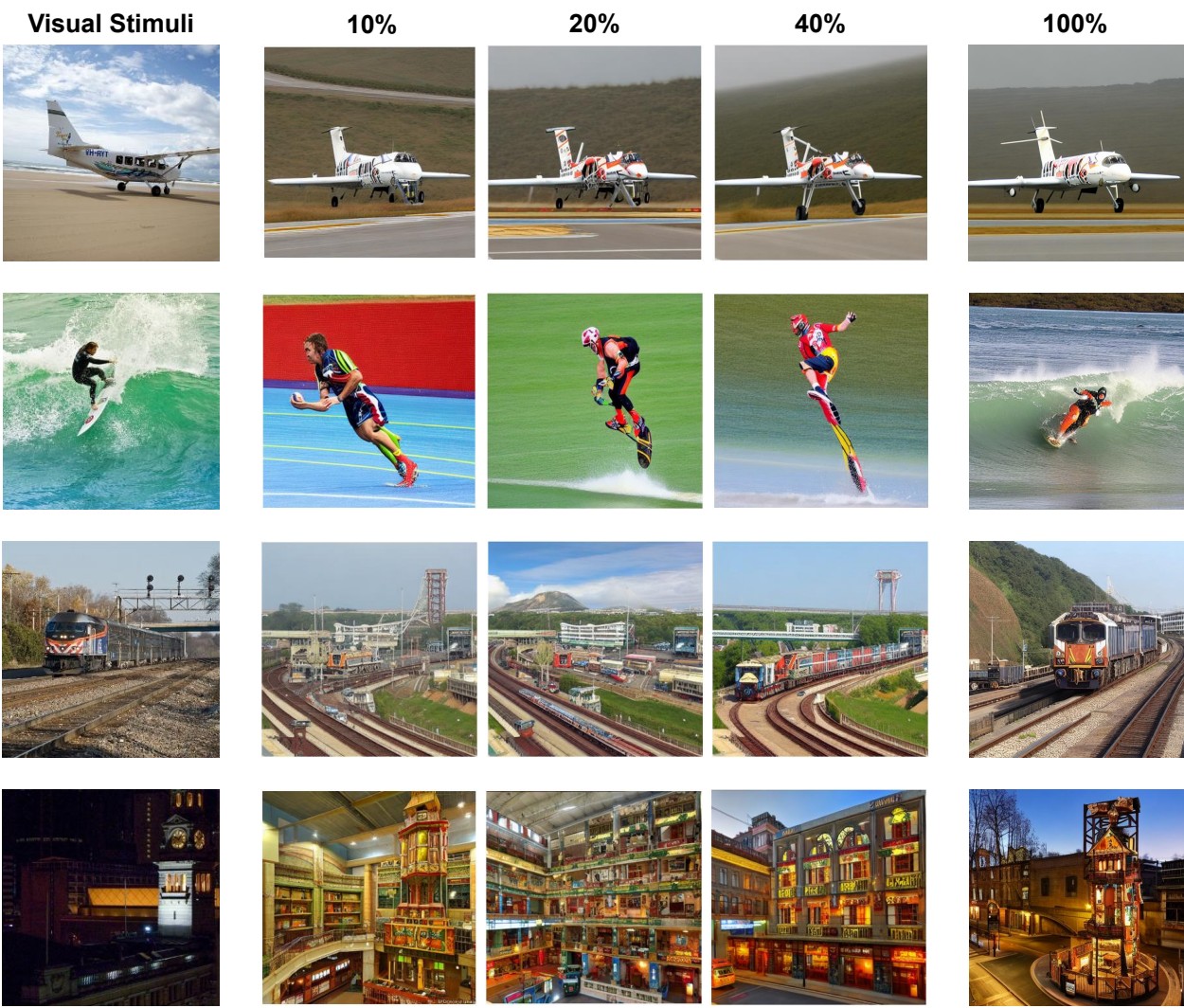

*Figure 20.* Few-shot brain reconstruction results on Subj1. Compared to full training (100%), our method can also roughly decode the brain signals with limited data. No more then 40%, MindCustomer is already capable of precise reconstruction.

