# OpenReview forum: "MindCustomer: Multi-Context Image Generation Blended with Brain Signal"
_ICML.cc/2025/Conference — ICML 2025 poster_

### Official Review · Reviewer_oSfB · 2025-03-11

**Overall Recommendation:** 3

**Summary:**

This paper introduces MindCustomer, a novel multi-context image generation framework that integrates visual brain signals with traditional image and text modalities to enable brain-controlled image customization. The authors propose a method to blend brain semantics into the image generation process, addressing key challenges such as multi-modal integration, and context preservation.

**Claims And Evidence:**

1.  As the first work,MindCustomer successfully implements multi context blended with brainsignal in image generation. （On the right side of line 70）

Evidence:The paper provides a thorough literature review, highlighting that previous works have focused on text-to-image generation or brain signal reconstruction but not on using brain signals to guide multi-context image generation.

2. It achieves high-quality image customization guided by cross-subject brain contexts.（On the right side of line 72）
Evidence: The paper presents qualitative results showing high-quality image outputs that faithfully represent the provided contexts.

Quantitative metrics (e.g., CLIP-I, DINOv2, CLIP-IQA) demonstrate superior performance compared to baseline methods. User studies also indicate a preference for MindCustomer's results over baselines.

3. We propose the Image-BrainTranslator(IBT) for cross-subject brain data augmentation, ensuring stable embeddings.（On the right side of line 104）

Evidence: The authors demonstrate the effectiveness of IBT through experiments showing that it can simulate brain signals from visual stimuli with moderate to high correlation to ground-truth fMRI data. Additionally, they show that IBT-augmented data improves the stability of brain embeddings.

**Essential References Not Discussed:**

See "Relation To Broader Scientific Literature".

**Experimental Designs Or Analyses:**

The authors compare MindCustomer to several ablation variants, including models without IBT, without fine-tuning and optimization, and without optimization. The results are evaluated using the same metrics and qualitative analysis as in the main experiments.

The effectiveness of the IBT may vary across subjects due to inherent differences in brain signal patterns. The paper could benefit from a more detailed analysis of how well the IBT generalizes across different subjects. And while the Pearson correlation measures similarity, it does not directly assess semantic alignment. Additional metrics or qualitative analysis could provide deeper insights into the semantic fidelity of the simulated brain signals.

While the ablation studies show improvements, they may not fully capture the interactions between components. More detailed analysis or additional ablation variants could provide deeper insights. For example, various indicators calculated from the reconstructed images of each part after ablation can be listed. The images displayed now are carefully selected and lack strong persuasiveness.

**Methods And Evaluation Criteria:**

Quantitative Metrics: The paper uses metrics such as CLIP-I, DINOv2, and CLIP-IQA to evaluate context similarity and image quality. These metrics are widely accepted in the field of image generation and are relevant for assessing the model's performance.

I have some concerns regarding the ablation experiments presented in Figure 8. The figure only provides two pairs of Image/Brain context examples and does not quantitatively measure the true quality of the proposed IBT, fine-tuning, and optimization methods in this paper.

The results in Table 4 appear to indicate that the fMRI image reconstruction framework proposed in this paper underperforms compared to MindBridge and UMBRAE, suggesting that there is no actual performance improvement in this study.

**Other Comments Or Suggestions:**

After passing through Semantic Extractor, the figure points to the loss function. Suggest modifying the drawing to make it clearer.

The Method section of the article, especially the definition of variables in Brain Representation Pre training, is quite confusing. Although the author's meaning can be understood, it is difficult to read. I think it's because some variables were not used or further explained after being defined, such as EIa and eIa.

**Other Strengths And Weaknesses:**

The experiments in the article mainly focus on qualitative analysis, with too little emphasis on quantitative analysis. There is too little comparison with other models to fully demonstrate its effectiveness. It is strongly recommended to add quantitative indicators to key experiments such as ablation experiments, comparative experiments with image fusion, and generalization experiments. At the same time, some calculations of reconstructed image indicators can be added.

This paper proposes an Image/Brain context-guided image generation model that requires both Image and fMRI inputs during the inference process. Additionally, the textual inversion technology is not new and has been widely used. I believe that using a method similar to MindEye's [1] prior diffusion and SDXL for image generation, followed by optimization of image features, would not be inferior to the approach presented in this paper. Therefore, I am not convinced of the superiority of the proposed method over other existing methods.

References

[1] Scotti P, Banerjee A, Goode J, et al. Reconstructing the mind's eye: fmri-to-image with contrastive learning and diffusion priors[J]. Advances in Neural Information Processing Systems, 2023, 36: 24705-24728.

**Questions For Authors:**

1. Is the encoding model for predicting fMRI from images trained separately for each subject?

2. The article mentions the use of shallow subject-wise embedders  and deep shared embedders  to obtain two different types of embeddings. Can each embedder obtain a different type of embedding? Are they the same thing as the Shallow Layer and Shared Deep Layer in Figure 2 (1)? If so, there seems to be a sequential relationship, so how can two different types of embeddings be obtained from a shared deep layer? This part confuses me.

**Relation To Broader Scientific Literature:**

The Image/Brain context-guided image generation proposed in this paper does not represent a particularly novel technology and can be viewed as a combination of existing techniques (e.g., MindEye2 (ICML 2024) and MindArtist (CVPR 2024)). It is recommended that the authors objectively evaluate the performance of the proposed methodological framework and emphasize its distinctions from previous work. This would help to highlight the contributions of this paper.

**Theoretical Claims:**

The theoretical claims in the paper are primarily supported by empirical evidence rather than formal proofs. The authors provide convincing demonstrations of the effectiveness of their methods through experiments, comparisons, and user studies. However, there are potential areas for further theoretical analysis, such as:

Robustness of the IBT: Analyzing the theoretical guarantees and limitations of the IBT in simulating brain signals. The correlation metric alone may not fully capture the semantic alignment between simulated and real brain signals. Additionally, the effectiveness of the IBT may depend on the quality and diversity of the training data.

Generalization Bounds: Exploring theoretical bounds on the model's generalization ability, especially in the context of few-shot learning. The generalization ability is based on empirical results from a limited number of subjects. Theoretical analysis could further explore the conditions under which the model generalizes well and the potential limitations in more diverse populations.

---

> ### Author Rebuttal · Authors · 2025-03-29
>
> We sincerely thank you for your reviews!
>
> ***Rebuttal Link*** (https://anonymous.4open.science/r/ICML2025-Rebuttal-MindCustomer) is for supplemented results in rebuttal, please refer to it.
>
> **Q1: Quantitative measures of ablation study and indicators of reconstructed image**
>
> **In the submitted paper, we have presented the quantitative measures of our ablation study in the right half of Table 1**, please refer to it. Due to page limitations, we reasonably merged these results with the baseline comparison within one table and provided a table reference in the relevant text (line 437). Furthermore, **we have listed the indicators of the reconstructed images in the submitted paper, please refer to Table 4**.
>
> **Q2: Table 4 illustration**
>
> As the first work on multi-context (brain, image, text) blended generation, MindCustomer needs accurate context extraction. Table 4 demonstrates our method’s effectiveness in brain semantics extraction and its competitiveness with SOTA brain decoding methods. The main experiments in this paper show that our brain encoding, combined with the proposed effective cross-modal fusion pipeline, is sufficient to achieve high-quality blending. While existing fMRI reconstruction methods may offer better encoding, they introduce more conditions and complexity, making them inefficient in multi-context blended generation.
>
> **Q3: Further theoretical analysis**
>
> Since our proposed IBT is an MLP-based model with a relatively simple structure, we reduced some theoretical analysis in other aspects (e.g. robustness). Thank you for providing additional discussion ideas, which further enhance our theoretical analysis. We will explore the robustness of the IBT model across different data and individuals, as well as its generalization to data volume in the few-shot generation.
>
> **Q4: Metrics or qualitative analysis of semantic fidelity of the simulated brain signals**
>
> We have provided additional qualitative and quantitative analyses of IBT-simulated brain signals in **Figures 15 and 16 of the original Appendix**. Please refer to them.
>
> **Q5: Comparison with other methods**
>
> - We have described a detailed analysis and differences with references (e.g. MindEye2, MindArtist), which can be found in the paper’s introduction (Left: Line 66-109, Right: Line 55-69) and related work section (Left: Line 150-164, Right: Line 135-155). Please refer to them.
>
> - To illustrate MindCustomer's performance, we add another baseline per the review's suggestion: using MindEye for fMRI-to-image reconstruction, followed by Versatile Diffusion (a mature image-blending model). In the baseline, VD requires no extra finetuning or optimization since it's pre-trained on large text-image datasets. In contrast, MindCustomer's finetuning adapts brain representation to diffusion's latent embedding, while optimization addresses semantic overlap between image and brain contexts—**unique innovations of our method**. As shown in the ***Rebuttal Link*** (**Rebuttal Figure 3 & Table 2**), due to the brain interpretation deviation caused by fMRI reconstruction and some semantic conflict from dual contexts resulting from the blending diffusion, the baseline shows some semantic deviation and loss. Benefiting from our unique designs, MindCustomer generates high-quality results. **We also explained this in the submitted paper (Experimental analysis: line 350-357 (right), Intro: line 091-095 & 058-065 (right)), please refer to them.** We will add the comparison results to better demonstrate MindCustomer's performance.
>
> **Q6: Clarity**
>
> Thank you for the suggestion. We will revise the output of the Semantic Extractor in the figure clearly. It should add the output embedding of the Semantic Extractor with the ground truth to construct the loss for model training. Besides, in the section of Brain Representation Pre-training, we will reduce some unnecessary symbols to make it simpler and easier to understand.
>
> **Q7: Is the encoding model for predicting fMRI from images trained separately for each subject?**
>
> IBT is subject-wise, please refer to line 181 (right). Due to the significant differences in original brain signals from different subjects, we designed the IBT for each subject. Experiments **(e.g. Figure 15 & 16, and Table 6 in Appendix)** show that even when IBT is lightweight (several layers of MLP), it can still effectively fit each subject's brain signals. Therefore, the subject-wise IBT does not increase the complexity within an acceptable range.
>
> **Q8: Can each embedder obtain a different type of embedding?**
>
> In line 203 (right): "We employ the shallow subjectwise embedders $E_{∫}$ and deep shared embedders $E_{⌈}$ to obtain the two types of embeddings of voxels", the two types refer to brain signal encodings supervised by CLIP-encoded **text** and **image**, respectively, as described in line 199 (right), rather than different types of encodings from different embedders. We will revise the description here for clarity.

---

### Official Review · Reviewer_2Q5v · 2025-03-12

**Overall Recommendation:** 3

**Summary:**

This paper proposes a new task: image customization with brain signals, which is novel and interesting. The authors introduce the image-brain translator and brain embedded to align various modalities, including images, fMRI, and texts, together for generating new images with a versatile diffusion model.

**Claims And Evidence:**

Claims are reasonable and supported by references and evidence.

Though, one reference mistake on ln139: "Additionally, Takagi. etc (Takagi & Nishimoto, 2023) employs Masked Autoencoders (MAE) to improve the encoding of brain signals, enabling more accurate reconstructions." Takagi & Nishimoto did not use MAE to encode brain signals in the mentioned reference. Please double-check the reference.

**Essential References Not Discussed:**

N/A

**Experimental Designs Or Analyses:**

Experimental designs and analysis make sense.

**Methods And Evaluation Criteria:**

The methods and evaluation criteria make sense.

**Other Comments Or Suggestions:**

Details of the user study should be introduced, such as study protocols, demographic information, etc.

**Other Strengths And Weaknesses:**

I think the quality of this paper is good in general. It presents an interesting new task with careful design for multimodal alignment. However, I have the following concerns.

- On ln185, the authors mentioned using adaptive max pooling to achieve the subject-invariant voxel sizes. However, various subjects have quite different activation regions even when viewing the same stimuli. Using adaptive max pooling as mentioned would lose or change the original position information of voxels. Therefore, this choice should be further discussed and justified.

- The baseline in Section 4.2 is not convincing enough. Ideally, the comparison should be between another fMRI blending model, rather than image bleeding models (no matter whether it is mask-free or signle-image generation or not). I would suggest using the output of an fMRI-to-image model as the input of another image-blending model as a baseline.

**Questions For Authors:**

See the previous sections.

**Relation To Broader Scientific Literature:**

The proposed task is new in this area, which is interesting and could have an impact on the future application of the brain-computer interface.

**Theoretical Claims:**

N/A

---

> ### Author Rebuttal · Authors · 2025-03-28
>
> We sincerely thank you for your reviews!
>
> ***Rebuttal Link*** (https://anonymous.4open.science/r/ICML2025-Rebuttal-MindCustomer) is for supplemented results in rebuttal, please refer to it.
>
> **Claims And Evidence: Reference error**
>
> Thank you for pointing out the citation error. It should be: Takagi et al.[1] reconstruct high-resolution images from brain activity while preserving rich semantic details, without the need for training or fine-tuning complex deep generative models. Mind-Vis[2] utilizes Masked Autoencoders (MAE) to enhance brain signal encoding, enabling more accurate reconstructions. We will correct this citation error.
>
> [1] High-resolution image reconstruction with latent diffusion models from human brain activity
>
> [2] Seeing Beyond the Brain: Conditional Diffusion Model with Sparse Masked Modeling for Vision Decoding
>
> **Other Strengths And Weaknesses1: Choice of adaptive max pooling**
>
> - Beyond unifying brain signals across subjects in scales, we believe adaptive max pooling helps mitigate the issue of sparse brain signal sampling, preserving critical semantic information. Additionally, reference MindBridge[3] shows that max-pooling provides better brain signal aggregation compared to average pooling or interpolation. We will further clarify this in our paper.
>
> - Besides, in the discussion and future work sections, we will acknowledge that this is not necessarily the optimal approach. In neuroscience, effective brain signal sampling requires a comprehensive analysis of cross-subject variations in numerical values, regions, and data quantity. We will explore the strengths and limitations of different sampling methods and provide further insights. We see this as a valuable direction for future work to design more effective sampling strategies.
>
> [3] MindBridge: A Cross-Subject Brain Decoding Framework
>
> **Other Strengths And Weaknesses2: The reason for original baseline design and results of recommended baseline**
> - The reason for us to choose the visual stimuli as the input rather than the fMRI-reconstructed image is that the images reconstructed via fMRI-to-image methods generally have lower semantic quality than the original visual stimuli. Then, using reconstructed images as inputs for blending would lead to weaker performance. Therefore, we chose to use the **higher-quality original visual stimuli** as the fusion input and set this as **a stronger baseline in our paper** to better demonstrate the effectiveness of our method.
>
> - Additionally, we also illustrate comparison results for using the output of a SOTA fMRI-to-image model (MindEye[4]) as input for another mature image-blending model (Versatile Diffusion[5]) as a baseline (advised in review). As shown in the ***Rebuttal Link*** (**Rebuttal Figure 3 & Table 2**), we conduct both qualitative and quantitative comparisons:  Due to the brain signal interpretation error caused by fMRI reconstruction and some semantic overlap between dual image contexts resulting from the blending diffusion model, this baseline shows some semantic deviation and loss. In contrast, the proposed **effective cross-modal fusing pipeline with the designed IBT model** addresses these issues. MindCustomer not only successfully blends different contexts in nature but also preserves semantics, thus generating high-quality results. **We also explained this in the submitted paper (Experimental analysis: line 350-357 (right), Intro: line 091-095 & 058-065 (right)), please refer to them.** We will also further supplement the above results and emphasize the analysis in the final version.
>
> [4] Reconstructing the Mind's Eye: fMRI-to-Image with Contrastive Learning and Diffusion Priors
>
> [5] Versatile Diffusion: Text, Images and Variations All in One Diffusion Model
>
> **Others: More details of the user study**
>
> In addition to the description of the User Study in the paper, we will provide more details about the study design in the Appendix. We randomly selected 110 pairs of comparison images, including ours and the baseline, and divided them into 11 groups. A total of 22 participants were involved, and to minimize bias caused by individual preferences, we randomly assigned every two participants to the same group, ensuring that each image received ratings from two different individuals. By comparing ours with the baseline, participants are required to score on each generated image based on the quality and consistency from 1 to 3, higher is better. Ultimately, we collected 220 answers for the 110 randomly selected image pairs from the 22 participants. The results broadly reflect the participants' consistent preference for our approach.

---

### Official Review · Reviewer_FgVa · 2025-03-14

**Overall Recommendation:** 2

**Summary:**

This paper proposes a novel framework, MindCustomer, to explore the blending of visual brain signals in multi-context image generation. This approach enables cross-subject generation, delivering unified, high-quality,  and natural image generation.

**Claims And Evidence:**

1. This paper claims some potential for practical application. But there are no related applications introduced. It will be better that authors are able to further illustrate some specific real-world applications.

2. Part of the contribution in this paper is summarized as the introduction of Image-Brain Translator (IBT). And authors provide some qualitative and quantitative results. But there are missing comparisons with other strategies handling the multi-identities generality such as MindEye2.

Reference:
MindEye2: Shared-Subject Models Enable fMRI-To-Image With 1 Hour of Data

**Essential References Not Discussed:**

It will be better if authors can also include the MindEye series of work.

MindEye2: Shared-Subject Models Enable fMRI-To-Image With 1 Hour of Data
MindEye1: Reconstructing the Mind’s Eye: fMRI-to-Image with Contrastive Learning and Diffusion Prior

**Experimental Designs Or Analyses:**

1. About the figure 6, why is the proposed approach able to achieve superior performance over the baseline? Theoretically speaking, the image signal is supposed to capture more visualization information than the brain signal.

2. For the brain decoding, how is performance compared with alternative designs such as MindEye1 ?


MindEye1: Reconstructing the Mind’s Eye: fMRI-to-Image with Contrastive Learning and Diffusion Prior

**Methods And Evaluation Criteria:**

The proposed approach is intuitive but the overall methodology is kind of combinational and has limited novelty. The evaluation criteria follows previous work and is reasonable.

**Other Comments Or Suggestions:**

N.A

**Other Strengths And Weaknesses:**

N.A

**Questions For Authors:**

Please kindly resolve the issues in Claims and Evidence sections.

**Relation To Broader Scientific Literature:**

It is a cross-domain work that combines both brain-to-image study and image generation field research. Both fields attract significant attention in the research field.

**Theoretical Claims:**

There are no theoretical claims involved in this paper.

---

> ### Author Rebuttal · Authors · 2025-03-28
>
> We sincerely thank you for your reviews!
>
> ***Rebuttal Link*** (https://anonymous.4open.science/r/ICML2025-Rebuttal-MindCustomer) is for supplemented results in rebuttal, please refer to it.
>
> **Q1: Claim of "potential for practical application"**
>
> - **As the first** to propose and achieve the fusion of fMRI, image, and text for customized creation, MindCustomer has demonstrated success on **the large-scale brain dataset NSD**.
>
> - We have discussed and explained this claim of potential application in detail in our **Discussion and Limitation section (Appendix D)**, please refer to it. **Constrained by the types of existing brain signal datasets**, we think that MindCustomer presents the potential application **in future real-world scenarios**.
>
> - Additionally, although constrained by the current types of brain signal data, we have also explored **few-shot generation for new subject in this paper**, simulating the effect of personalized generation when there is a limited amount of brain data in real-world scenarios.
>
> In all, we believe that MindCustomer is an important foundational exploration, and in the future, more real-world data collection will be involved to promote the application of MindCustomer.
>
> **Q2: Reference to MindEye**
>
> - Regarding the performance of the fMRI reconstruction task, it is slightly lower compared to MindEye. We believe the reasons lie in the fact that the MindEye series used four more subjects' data for training (MindEye2) and employed additional information, such as retrieval, brain captions, and more diffusion models, to better achieve brain reconstruction tasks. In contrast, **for MindCustomer's multi-context blending task, this additional information was not needed**. As such, a direct comparison between the two **may not be fair**. Meanwhile, we **have cited and analyzed a series of brain reconstruction works, including MindEye, in our paper (line 71 in intro and line 164 in related work)**. We not only acknowledge and appreciate their contributions to brain signal decoding but also analyze the differences between our work and these previous studies in terms of tasks and challenges. Unlike prior works that focus on better decoding brain signals themselves, the core of our work lies in how to fuse brain signals with various cross-modal information to create naturally composed images. **Besides, we will also include more performance discussion of the MindEye series of work in the final version to highlight respective advantages.**
>
> - Nevertheless, we would like to emphasize that, firstly, **we have fairly compared MindCustomer with several SOTA methods in the brain reconstruction field**. MindCustomer demonstrated strong competitiveness. Secondly, as for the task and challenges in this paper, it is not only to have a model with a certain level of brain semantic encoding ability but **more importantly**, how to integrate information from different modalities to generate natural images that preserve semantics. Our experimental results have proven that MindCustomer **is capable of effectively extracting brain signal semantics**, and **more importantly, it has demonstrated excellent performance in the personalized blending task**. Therefore, we believe that MindEye indeed makes an important contribution to the field of fMRI reconstruction, while MindCustomer also has its own merits in personalized multimodal fusion creation. We will further clarify the relation in final version.
>
> **Q3：Figure 6 question**
> - Although images may contain richer semantics, we observe that inputting two images into versatile diffusion (VD), a mature image fusion method for blended generation, often causes **semantic overlap**, making it hard to preserve distinct semantics. To address this, we designed VD fine-tuning based on image context to learn the image semantics, and more importantly, we freeze the fine-tuned VD model, and lightly optimize brain contexts with the target of image context. This helps mitigate semantic conflicts between different image and brain contexts. During the above process, we further utilize the proposed IBT to transfer image context to brain-embedding space to reduce the gap between modals. Thus, **the above-designed brain-blended cross-modal fusion** allows the final multimodal blending to better preserve each modality's semantics and produce more accurate, high-quality results. We also have explained this in the paper **(line 353, right)**: “This is likely ... ”, please refer to it.
>
> - Therefore, as for the method design of MindCustomer, we **innovatively propose IBT to simulate the process of visual stimuli being transformed into brain signals**. This not only expands brain signal data and enhances encoding capability but also **provides the above solution of brain-blended cross-modal fusion**. Overall, MindCustomer introduces well-designed modules and a structured pipeline tailored to the task, making it a simple yet effective approach rather than a naive combination.

---

### Official Review · Reviewer_4Rs7 · 2025-03-14

**Overall Recommendation:** 3

**Summary:**

This paper proposes MindCustomer, an image generation method with input of image, text and brain signals. The brain signals is converted by Image-Brain Translator(IBT) into image embeding space for subsequent text-image joint generation. The diffusion model is finetuned and the embeddings is optimized to achieve more plausible visual results.

**Claims And Evidence:**

yes

**Essential References Not Discussed:**

.

**Experimental Designs Or Analyses:**

yes

**Methods And Evaluation Criteria:**

yes

**Other Comments Or Suggestions:**

Do you have more visual results with only different text descriptions?

**Other Strengths And Weaknesses:**

## Strengths

1. The integration of brain signals for image editing shows successful results, surpassing baseline methods.


## Weaknesses


1. The contribution is limited to trivial fine-tuning of Versatile Diffusion and linear interpolation of two embeddings. The brain representation pre-training (IBT) and new-subject few-shot generation are largely derived from MindBridge, making the paper limited in technical contribution.
2. The introduction of ClipCap in IBT is not validated in ablation studies, making its actual contribution unclear.
3. Clarity issues in the description:
* The term "$\eta$" in equation (1) is not defined.
* The term "$e_p$" is not explained in equation (8).

**Questions For Authors:**

Please refer to weakness.

**Relation To Broader Scientific Literature:**

The paper builds upon previous work in brain decoding and image generation, particularly leveraging advancements in models like MindBridge.

**Theoretical Claims:**

yes

---

> ### Author Rebuttal · Authors · 2025-03-28
>
> We sincerely thank you for your reviews!
>
> ***Rebuttal Link*** (https://anonymous.4open.science/r/ICML2025-Rebuttal-MindCustomer) is for supplemented results in rebuttal, please refer to it.
>
> **W1: Comments on contributions**
> - Diffusion model fine-tuning and linear interpolation are well-established techniques widely used in many studies. While we also employ these techniques in our work, they are **not** the core contributions we claim in this paper.
>
> - Our main contributions first lie in **as the first work, MindCustomer proposed and successfully achieved the personalized image creation task by integrating brain signals with other traditional modalities**.
>
> - Second, we need to clarify that the designed **IBT, a model that simulates pseudo-brain signals from input images (please note that it is not a brain signal encoding model described in the review**: "The brain signals is converted by Image-Brain Translator(IBT) into image embeding space for subsequent text-image joint generation", "The brain representation pre-training (IBT)", "The introduction of ClipCap in IBT"). **IBT simulates brain signals from images to enhance subsequent brain signal semantic encoding and reduce the modal gap in the following cross-modal fusion pipeline, which is independent of MindBridge**. Specifically, based on the proposed IBT, we simulate subject-wise fMRI for each visual stimulus. In this way, we can augment the fMRI training dataset due to the stimulus for each subject is different from the original dataset. Then during brain representation pre-training, we simultaneously feed different subjects' fMRI of one stimulus for shared learning in one encoding model. **The above process is different from MindBridge**.
>
> - Third, more importantly, we propose **an effective cross-modal information fusion pipeline**. In order to blend image context with brain context, we designed VD fine-tuning based on image context to learn the image semantics, and then we froze the fine-tuned VD model and lightly optimized brain contexts with the target of image context. **This helps mitigate semantic conflicts between different contexts**. Besides, during the above process, we further utilize the proposed IBT and brain embedding model to transfer image context to brain-embedding space to **reduce the gap between different modals**. Thus, the above-designed brain-blended cross-modal fusion enables final natural integration and semantic preservation across modalities, producing natural and high-quality blending results.
>
> The above clarification can also be referred to the submitted paper (line 96, left - line 68, right). Therefore, **it is vital to highlight that these are our core contributions**, which fundamentally exceed simple VD fine-tuning or linear interpolation.
>
> **W2: Ablation study of ClipCap**
>
> In the ***Rebuttal Link*** (**Rebuttal Figure 1 & Table 1**), we include an ablation study of ClipCap. The blending results without ClipCap show lower semantic details with complete MindCustomer. Both the qualitative and quantitative results demonstrate the role of the introduced ClipCap in enhanced brain semantic extraction. Adding this experiment makes our study more comprehensive. Additionally, please note that **we have also conducted detailed numerical and visual ablation studies on the proposed *core* techniques (e.g. IBT, cross-modal information fusion pipeline) in the submitted paper**.
>
> **W3: Clarity**
>
> - $\eta$ in $G_{\eta}$ refers to the parameters of the IBT model $G$. Thank you for pointing out. We will add its definition.
> - **We have defined $e_{p}$ in the paper (line 256) before using it in equation (8)**,  "We feed the transferred image context $Bp$ into brain embedder $\epsilon$ to obtain the embeddings $e_{p}$." We will also appropriately add a further description of it in the subsequent text.
>
> **Others: More results with only different text descriptions**
>
> In the ***Rebuttal Link*** (**Rebuttal Figure 2**), we present more visual results with only different text contexts. As we can see, MindCustomer robustly creates multi-context images that are content-consistent and naturally integrated.

---

> > ### Comment · Reviewer_4Rs7 · 2025-04-06
> >
> > Thank the authors for the response, which addresses most of my concerns and clarifies the technical contributions. I have therefore decided to raise my score to 3.
> >
> > That said, I suggest the authors incorporate the key clarifications from the rebuttal into the revised version to make the paper clearer and more convincing to readers.

---

> > > ### Author Response · Authors · 2025-04-07
> > >
> > > Dear Reviewer 4Rs7,
> > >
> > > We sincerely appreciate your recognition of our work, and we're pleased that your concerns have been resolved! Based on the rebuttal, we will make further clarifications in the revised version.

---

### Decision · Program_Chairs · 2025-05-01

**Decision:**

Accept (poster)

**Comment:**

The paper received mixed ratings from four reviewers. Three of them had positive comments on the technical novelty and the experimental verifications. One reviewer provided a rating of 'weak reject', and had questions only regarding some elaboration on the experimental results and discussion. In the rebuttal, the authors clearly echoed those questions with detailed information. Based on all these comments and the rebuttal, AC decided to recommend a weak acceptance for this submission.